# Local and global influences on protein turnover in neurons and glia

**Aline R Dörrbaum[1,2], Lisa Kochen[1], Julian D Langer[1,3]\*, Erin M Schuman[1]\***

[1]Max Planck Institute for Brain Research, Frankfurt, Germany; [2]Faculty of Biological Sciences, Goethe University Frankfurt, Frankfurt, Germany; [3]Max Planck Institute of Biophysics, Frankfurt, Germany

**Abstract** Regulation of protein turnover allows cells to react to their environment and maintain homeostasis. Proteins can show different turnover rates in different tissue, but little is known about protein turnover in different brain cell types. We used dynamic SILAC to determine half-lives of over 5100 proteins in rat primary hippocampal cultures as well as in neuron-enriched and glia-enriched cultures ranging from <1 to >20 days. In contrast to synaptic proteins, membrane proteins were relatively shorter-lived and mitochondrial proteins were longer-lived compared to the population. Half-lives also correlate with protein functions and the dynamics of the complexes they are incorporated in. Proteins in glia possessed shorter half-lives than the same proteins in neurons. The presence of glia sped up or slowed down the turnover of neuronal proteins. Our results demonstrate that both the cell-type of origin as well as the nature of the extracellular environment have potent influences on protein turnover.

DOI: https://doi.org/10.7554/eLife.34202.001

*For correspondence:
julian.langer@biophys.mpg.de
(JDL);
erin.schuman@brain.mpg.de
(EMS)

**Competing interests:** The authors declare that no competing interests exist.

## Introduction

Proteins, the fundamental units of all cells, exhibit dynamics in their expression levels in response to intracellular and extracellular signals. Protein turnover, measured in cells, is the net result of protein synthesis and degradation. Under steady-state conditions, proteins are continuously turned over (*Boisvert et al., 2012*; *Cambridge et al., 2011*; *Cohen et al., 2013*; *Price et al., 2010*). Protein turnover allows for the removal of damaged proteins and their replacement by new proteins. In addition, continuous protein turnover is required and exploited to enable cells to dynamically adjust their proteome according to internal and external perturbations and signals. Turnover rates have likely been optimized during evolution such that individual proteins possess a life time that represents the balance between energy-saving stability and dynamic flexibility. In the brain, proteome remodeling using protein synthesis and degradation is required for learning and memory formation (*Sutton and Schuman, 2006*). In addition, several forms of synaptic plasticity studied *in vitro* also require protein synthesis and protein degradation (*Ehlers, 2003*; *Kang and Schuman, 1996*; *Rosenberg et al., 2014*; *Schanzenbächer et al., 2016*). During homeostatic scaling of cultured hippocampal neurons, for example, specific sets of proteins show increased or decreased protein synthesis associated with the up- or downscaling of synapses (*Schanzenbächer et al., 2018*; *Schanzenbächer et al., 2016*). The turnover of brain proteins has been measured both *in vivo* and *in vitro*. *Price et al. (2010)* used *in vivo* metabolic [15]N-labelling and the subsequent mass spectrometric analysis of whole brain homogenates to derive the turnover rates for 1010 proteins (average half-life =~9 days) in the mouse brain. Another study used *in vitro* metabolic labeling of primary cortical cultures to measure relatively shorter half-lives for 2802 brain proteins (average half-life =~5 days; (*Cohen et al., 2013*)). In both of these studies, however, average half-lives were obtained using mixed cell populations including multiple neuronal and glial cell types. It is known that proteins can show very different turnover rates in different tissues (*Price et al., 2010*) or different cell types

(*Mathieson et al., 2018*) of the same organism, but little is known about protein turnover rates in different cell types of the brain.

To address these issues, we used a dynamic SILAC approach to determine protein half-lives in primary hippocampal cultures (containing a mixture of neurons and glia cells), as well as in neuron-enriched and glia-enriched cultures. Our results demonstrate that both the cell-type of origin as well as the nature of the extracellular environment have potent influences on protein turnover. In neurons, protein turnover rates are related to a protein's function as well as the intracellular environment and protein interactions.

## Results

### Determination of protein half-lives in primary hippocampal cultures

To determine protein half-lives, we used a dynamic stable isotope labeling with amino acids in cell culture (SILAC) approach in combination with LC-MS/MS analysis using mature primary hippocampal cultures (containing both neurons and glia, the latter of which do not undergo cell division, owing to the confluence and maturity of the cultures, see Materials and methods and *Figure 6—figure supplement 5*). Mature primary neuronal and glial cultures mimic the physiology of cells *in-vivo* and are an important system to study the molecular and cellular mechanisms that underlie the physiological and pathophysiological function of neuronal networks. Cultured hippocampal cells were maintained for 18–19 days in growth medium containing natural 'light' arginine and lysine, then the medium was exchanged to one containing 'heavy' isotopically labelled arginine and lysine (*Figure 1A*). To reduce cellular stress (see Materials and methods), we retained a small amount of the 'light' medium and an excess of 'heavy' medium was added, yielding a final 4:1 ratio of heavy to light amino acids. Note that following the medium switch, mostly 'heavy' arginine and lysine are incorporated into nascent proteins while the fraction of 'light' proteins decays over time. The cells were allowed to incorporate the 'heavy' amino acids for 1, 3 or 7 days before they were harvested, lysed and prepared for MS analyses (*Figure 1A*). An additional sample was harvested just before the medium change at t = 0. For all identified peptides, the fraction of heavy and light peptide signal was quantified by MS at each time point (see peptide.txt uploaded to PRIDE). As the medium contained residual 'light' amino acids, we developed a correction factor for each peptide using the

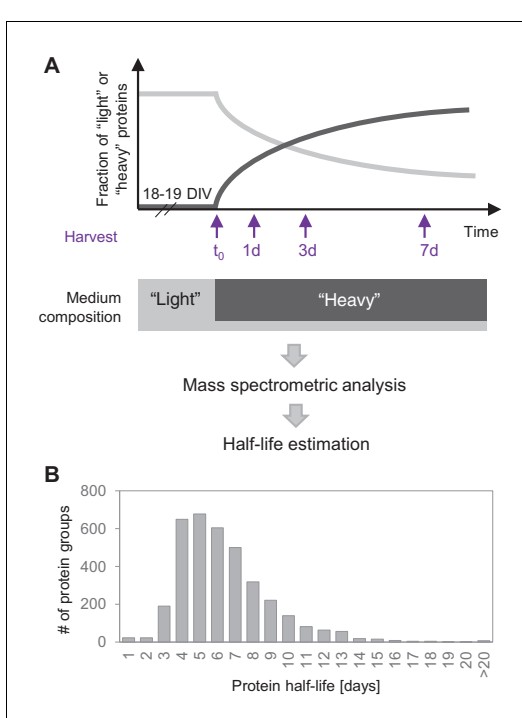

**Figure 1.** Protein turnover in neurons. (**A**) Protein half-lives were determined by a dynamic SILAC approach in combination with MS. Cells were grown for 18–19 days in medium containing natural amino acids ('light') and then switched to a medium containing heavy isotopically labelled arginine (R10) and lysine (K8). Upon the medium change, the 'heavy' amino acids were incorporated into newly synthesized proteins, whereas the fraction of 'light' pre-existing proteins decayed over time. The cells were harvested 1, 3 and 7 days after the medium change as well as just before the medium change ($t_0$) and the fractions of newly synthesized proteins and pre-existing proteins were determined by MS. Protein half-lives were determined based on first order exponential fitting of the fraction of pre-existing proteins over time. (**B**) Distribution of protein half-lives in primary hippocampal cultures ranged from <1 day to >20 days.

DOI: https://doi.org/10.7554/eLife.34202.002

The following figure supplements are available for figure 1:

**Figure supplement 1.** Data processing workflow of dynamic SILAC samples.
DOI: https://doi.org/10.7554/eLife.34202.003

**Figure supplement 2.** Accuracy of turnover estimation.
DOI: https://doi.org/10.7554/eLife.34202.004

**Figure supplement 3.** Comparison to *in vivo* study.
DOI: https://doi.org/10.7554/eLife.34202.005

**Figure supplement 4.** Protein decay clusters.
DOI: https://doi.org/10.7554/eLife.34202.006

probability of the incorporation of a 'light' amino acid into a nascent protein from peptides containing two arginines or lysines due to a missed tryptic cleavage site (*Figure 1—figure supplement 1*; see Materials and methods). Using the above approach, we determined the protein half-lives of 3610 protein groups, using a first order exponential fit of the pre-existing protein levels over time (*Figure 1A,B*; *Supplementary file 1*). The proteins we analyzed exhibited a range of half-lives from <1 day to >20 days; the median half-life was 5.4 days. For extremely short-lived proteins, where the data could not be accurately fit by an exponential decay curve, we assigned a half–life of <1 day. The distribution of protein half-lives, shown in *Figure 1B*, is right skewed, reflecting a long-tail of proteins with longer than average half-lives. To assess the quality of the dataset and the accuracy of the protein half-lives, we computed the standard error (SE) of the rate constant as well as the coefficient of determination ($R^2$) of the fit for each protein (*Figure 1—figure supplement 2A, D* and *Supplementary file 1*). Across the complete set of protein half-lives in mixed cultures, we obtained a mean SE of the rate constant of ~0.007 and a median $R^2$ of 0.96 demonstrating an accurate determination of protein half-lives.

## Validation of protein half-lives

To validate the half-lives obtained with our MS analysis, we used a different pulsed metabolic labelling technique, FUNCAT-PLA, that allows *in situ* visualization of newly synthesized proteins-of-interest using non-canonical amino acids like azidohomoalanine (AHA) and antibodies (*tom Dieck et al., 2015*). Extracellularly applied AHA crosses cell membranes and gets charged by endogenous methionyl tRNA synthetases and incorporated into nascent proteins (*Dieterich et al., 2006*). For this analysis, we chose four proteins (Bassoon, TrkB, GM130 and LaminB1) which exhibit different cellular functions and half-lives ranging from ~3 days to ~11 days based on our MS analyses. To analyze the half-lives of these proteins, a pulse of the non-canonical amino acid AHA was delivered for 2 hr to hippocampal cultures (DIV 18). Then, at variable intervals (a.k.a. chase), the pool of pulse AHA-labelled Bassoon, TrkB, GM130 or LaminB1 was tagged and visualized (*Figure 2A* and *tom Dieck et al., 2015*). Previous studies demonstrated that 2 hr AHA incorporation is not toxic to cells and does not affect global protein synthesis or degradation rates (*Dieterich et al., 2006*). In these experiments we labelled GM130 or LaminB1 and also used data obtained for TrkB and Bassoon from *tom Dieck et al., 2015* using identical labeling and tagging conditions. For both GM130 and LaminB1, proteins associated with Golgi and nuclear function, respectively, a decrement in the number of labelled puncta was observed over days, yielding measured half-lives of 1.9 and 4.4 days, respectively (*Figure 2B*, *Figure 2—source data 1*). For all four proteins examined, the half-lives obtained by the FUNCAT-PLA analysis were well-correlated with the half-lives obtained with MS (Pearson correlation = 0.983; *Figure 2C*). We noted, however, that the half-lives obtained with FUNCAT-PLA were systematically shorter. This could be owing to a reduced stability of the AHA-containing proteins or the azide group used for tagging or other factors, as discussed below.

## Protein localization and function

We next examined the relationship between a protein's half-life and its location and/or function in cells. We calculated the half-lives for proteins that reside in cytoplasm, nucleus or plasma membrane, as well as in many different organelles (*Figure 3—source data 1*). As shown in *Figure 3A*, many groups exhibited a wide range of protein half-lives that were not significantly different from the population average. An exception to this were proteins associated with the nucleus as well as several groups of membrane proteins, including those associated with the plasma membrane, the Golgi apparatus membrane and the ER membrane which all exhibited a shorter than average half-life. Mitochondrial proteins also deviated from the population average: they exhibited significantly longer half-lives than all other groups. This was a counter-intuitive observation given the exposure of mitochondrial proteins to elevated levels of reactive oxygen species (*Adam-Vizi and Chinopoulos, 2006*). Lastly, despite their remote location within axonal and dendritic arbors, we found that synaptic proteins as a group do not exhibit significantly different half-lives from the population (*Figure 4—source data 1*; see below and *Figure 4* for more discussion of individual synaptic protein half-lives).

We next grouped proteins by their half-lives, ascending from less than 3 days to greater than 14 days and performed a Gene Ontology (GO) analysis. We found that proteins associated with distinct cellular components (B) or molecular functions and biological processes (C) were significantly over-

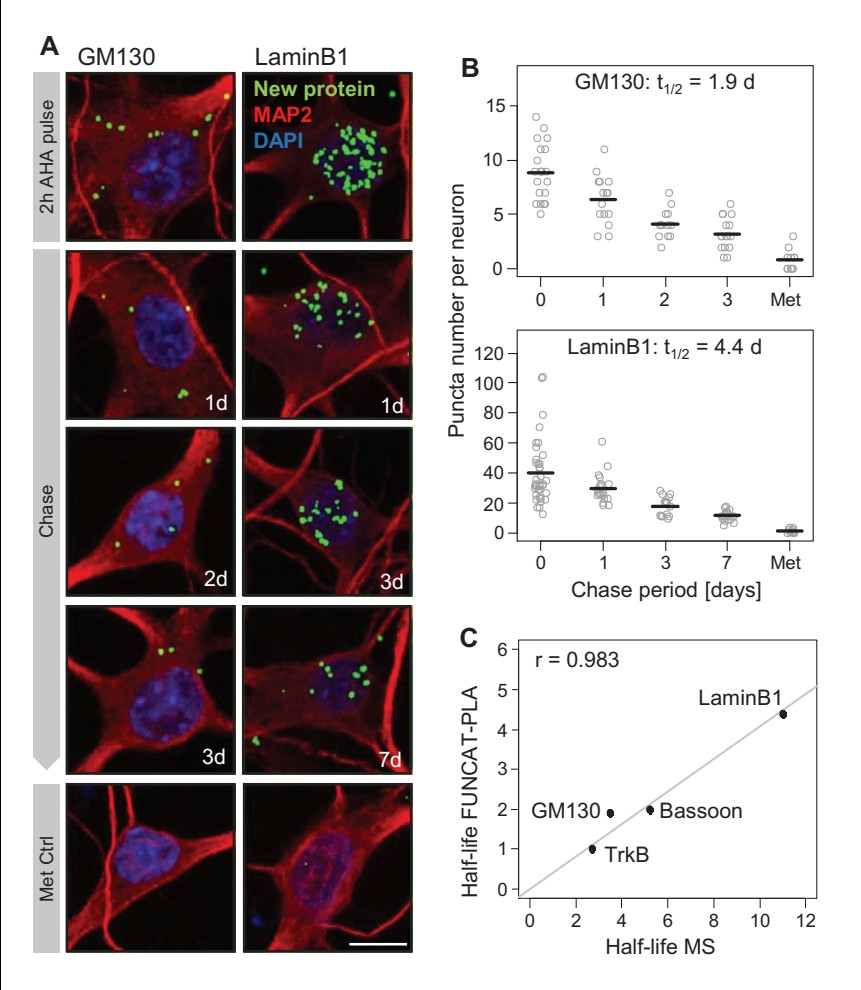

**Figure 2.** Validation of mass spectrometry results with metabolic labeling and visualization. Protein half-lives were determined by FUNCAT-PLA for GM130 and LaminB1. Degradation of newly synthesized proteins (following a 2 hr AHA pulse) was monitored for 0 (no chase), 1, 2 and 3 days for GM130 and for 0, 1, 3 and 7 days for LaminB1. (**A**) Representative images of neurons with newly synthesized LaminB1 and GM130 as indicated (green), cell body and dendrites (MAP2, red), nuclei (DAPI, blue). Puncta were dilated for visualization purposes only. Scale bar = 10 μm. (**B**) Group analysis of experiments shown in A. Puncta number per neuron is shown for individual cells (grey dots) as well as the mean (black line). Protein half-life was determined based on exponential fitting of the mean puncta number per cell. (**C**) Protein half-lives measured by FUNCAT-PLA and MS exhibit a high and significant correlation (Pearson correlation (r) = 0.983); data for TrkB and Bassoon were obtained from *tom Dieck et al., 2015*.
DOI: https://doi.org/10.7554/eLife.34202.007
The following source data is available for figure 2:

**Source data 1.** FUNCAT-PLA results.
DOI: https://doi.org/10.7554/eLife.34202.008

represented ($p<0.05$, Bonferroni corrected) in certain half-live groups (*Figure 3—source data 2*). For example, receptors, signaling molecules, cell adhesion molecules, and proteins involved in stimulus response and cell communication were over-represented in the segment of comparably short-lived (<3 days) proteins. Ribosomal proteins were enriched in a group with relatively long half-lives (6–11 days). Amongst the most long-lived (12–15 days) groups we found a significant over-representation of proteins involved in energy metabolism. In this long-lived group there was also an over-representation of histones and nuclear pore proteins, consistent with previous reports (*Savas et al., 2012*; *Toyama et al., 2013*). We analyzed in greater detail the proteins that inhabit the presynaptic and postsynaptic compartments. As mentioned above, synaptic protein show a wide range of

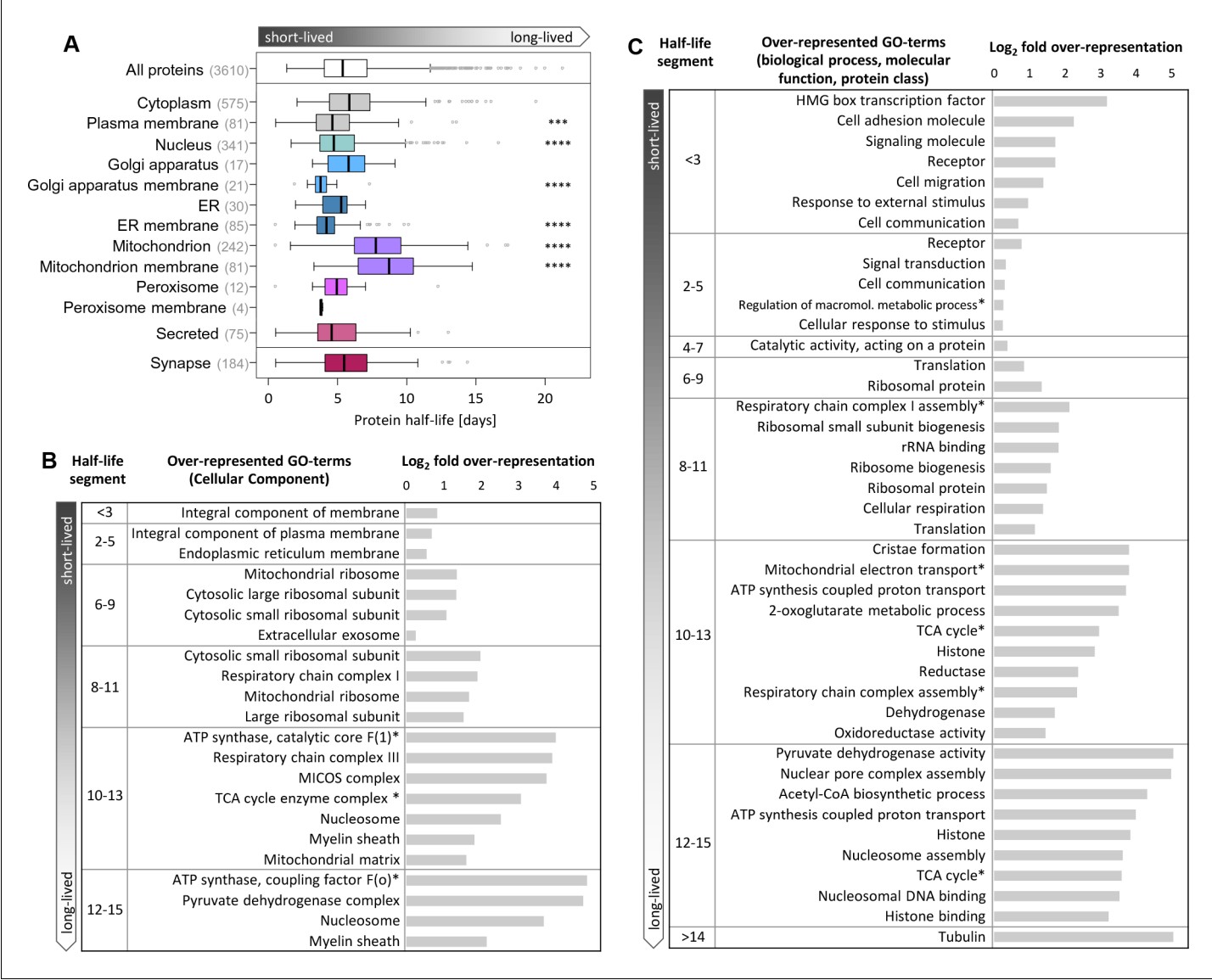

**Figure 3.** Half-lives of proteins with different cellular locations and functions. (**A**) Half-life distribution of proteins in different cellular compartments (LocTree annotation; score ≥50) and of synaptic proteins (extracted based on UniProt GO annotation). Number of protein groups assigned to each compartment is given in brackets. Mitochondrial proteins are significantly longer-lived compared to the complete set of quantified half-lives. Except in mitochondria, membrane proteins tend to be shorter-lived compared to soluble proteins in each compartment. Selected GO terms for cellular component (**B**) as well as biological process, molecular function and Panther protein class (**C**) that are over-represented (p-value<0.05) in distinct half-life segments. The complete set of over-represented GO terms and the full name of abbreviated GO terms (marked with *) are given in *Figure 3—source data 2*.

DOI: https://doi.org/10.7554/eLife.34202.009

The following source data and figure supplement are available for figure 3:

**Source data 1.** Half-lives of proteins at different sub-cellular localizations.
DOI: https://doi.org/10.7554/eLife.34202.011

**Source data 2.** GO analysis of half-life segments.
DOI: https://doi.org/10.7554/eLife.34202.012

**Figure supplement 1.** N-terminal sequence analysis.
DOI: https://doi.org/10.7554/eLife.34202.010

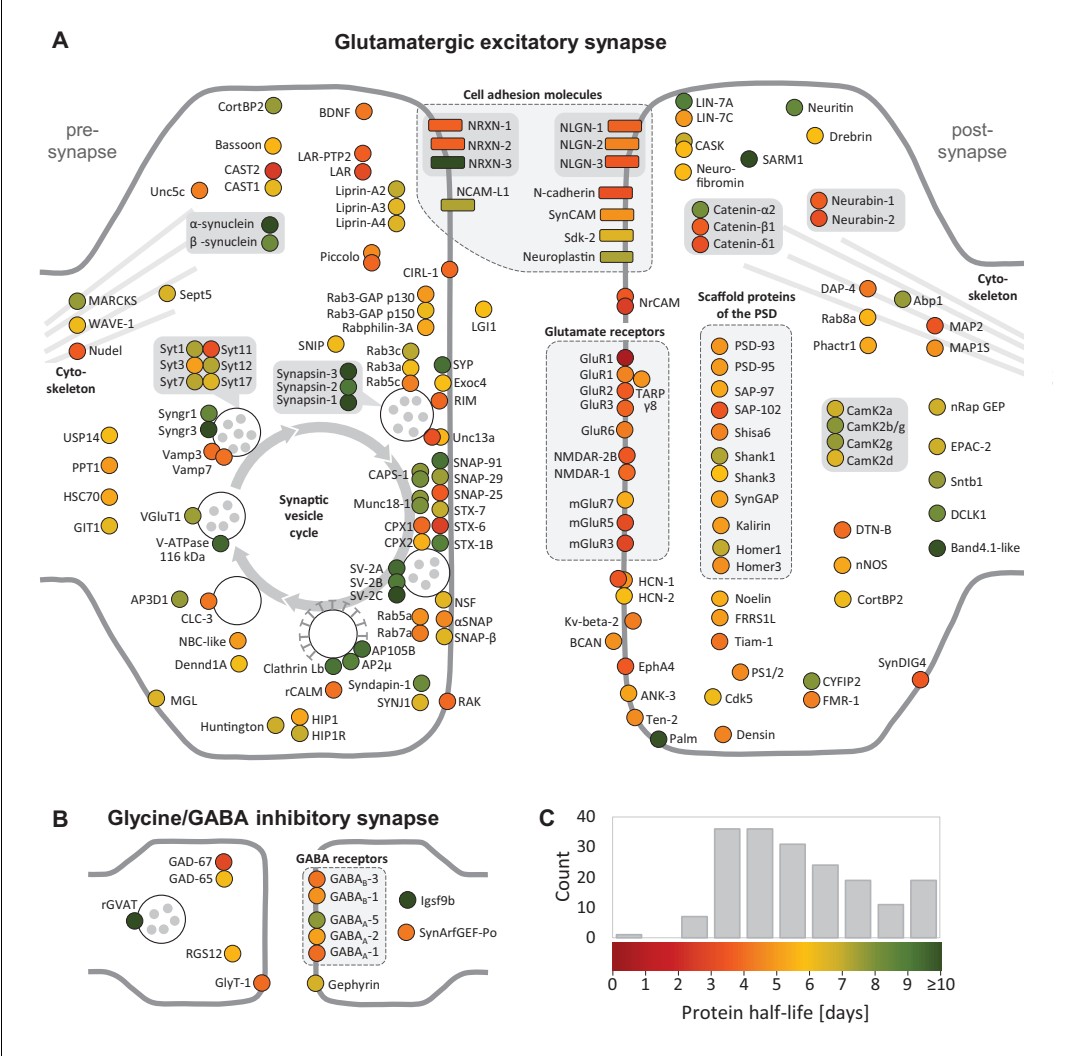

**Figure 4.** Half-lives of synaptic proteins. Proteins located at glutamatergic synapses (**A**) and glycinergic or GABAergic synapses (**B**). Half-lives are indicated by a color code from red (short-lived) to green (long-lived). Multiple circles for the same protein name show different isoforms. Interacting proteins are displayed in closer proximity. Proteins with similar functions are grouped together and the synaptic vesicle cycle is indicated by arrows. Abbreviated protein names are explained in *Figure 4—source data 1*.

DOI: https://doi.org/10.7554/eLife.34202.013

The following source data is available for figure 4:

**Source data 1.** Half-lives of synaptic proteins.

DOI: https://doi.org/10.7554/eLife.34202.014

protein half-lives, which are not significantly different from non-synaptic proteins. *Figure 4* depicts half-lives of all synaptic proteins in our dataset. We note that the receptors for the excitatory neurotransmitter glutamate (GluR1, GluR2, GluR3, GluR6, NMDAR1, NMDAR1B, mGluR3, mGluR5, and mGlusR7) exhibit shorter half-lives than the global protein population average (*Figure 4A*) while receptors for the inhibitory neurotransmitter GABA exhibited half-lives similar to the population average (*Figure 4B*). Relatively short half-lives were also observed for most of the cell adhesion molecules we identified, including most neuroligins (NLGN) and neurexins (NRXN) as well as N-cadherin. More long-lived proteins were found in the pre-synaptic compartment. Amongst the most long-lived synaptic molecules are those associated with synaptic vesicles and vesicle processing including the synapsins, the synaptic vesicle transporter family Sv2 as well as the Munc proteins (*Figure 4A*). We also noted in some cases that individual protein isoforms as well as members within a protein family

could exhibit very different half-lives, e.g. the synaptotagmins (Syt), the syntaxins (STX) and even CamK2 subunits (*Figure 4A*).

## Protein complex members

Proteins are functionally defined by their interactions and many proteins form stable multi-protein complexes to execute their cellular functions; these macromolecular complexes are now intensely studied both structurally and functionally (*Purdy et al., 2014*; *Wohlgemuth et al., 2015*). As such, we also investigated the half-lives of proteins in the context of the complexes they are associated with. We examined 314 protein complexes (composed of ≥5 proteins) and found that on average, proteins that are co-members of a complex have more similar half-lives than randomly selected proteins (*Figure 5A*; *Figure 5—source data 1*). We determined whether the protein half-lives of complex members are related to the size of the complex (in terms of number of protein constituents). We found that the variability of protein half-lives within complexes does not correlate with the number of constituents in a complex (*Figure 5—figure supplement 1*).

We next asked if the half-lives of multi-protein complex constituents correlate with the assembly and disassembly dynamics of the complexes. We analyzed in greater detail several protein complexes with different assembly and disassembly dynamics, including the spliceosome, the ribosome, the nuclear pore complex, and the ATP synthase (*Figure 5B–E*; *Figure 5—source data 2*). The spliceosome is a highly dynamic complex that assembles for each splicing event and then disassembles again afterwards. For the analysis of the spliceosome, we chose the human 'B-complex', the pre-catalytic complex primed for activation, which had recently been described structurally (pdb 5O9Z; *Bertram et al., 2017*). This structure contains 38 subunits, of which we detected 20 (*Figure 5B,F*). We found that these proteins exhibit a mean turnover of ~4.7 days, which is faster than the global population of proteins we measured. The ribosome and the ATP-synthase are thought to be more static complexes. Ribosomal proteins are synthesized in the cytoplasm and subsequently assembled into the large and small ribosomal subunits in the nucleus and nucleolus, where they interact with a variety of ribosomal assembly proteins and ribosomal RNA (*Peña et al., 2017*), before they are released back into the cytoplasm where they drive protein synthesis. Depending on the translational activity, the small and large ribosomal subunits can associate and dissociate, but the subunits themselves are considered to be stable throughout their life-time. The ribosome comprises ~79 proteins, 47 in the large subunit and 32 in the small subunit. We analyzed our data using a cryo-EM structure of the *Sus scrofa* ribosome (pdb 3J7R; *Voorhees et al., 2014*). We obtained half-lives for 70 of the 76 ribosomal proteins annotated in the structure, which exhibited a mean half-life of 7.8 days across all subunits (*Figure 5C,F*). Especially short-lived ribosomal proteins included proteins associated with both the large (e.g. Rpl21: 4.4 days) and small (e.g. Rps30: 4.0 days) subunits; we note that these proteins have not been recognized for extra-ribosomal functions (*Warner and McIntosh, 2009*). The ATP synthase assembles in a step-wise manner. The membrane-embedded rotor as well as the soluble stalk and head domains are produced and assembled independently, and full ATP synthase assembly is a finely-tuned process dependent on the energy demand of the cell (*Kucharczyk et al., 2009*). For the ATP synthase, we analyzed a recently published structure (pdb 5LQX; *Vinothkumar et al., 2016*) and detected 8 of the 12 ATP synthase proteins found in the structure (*Figure 5E,F*). The protein subunits exhibit a relatively limited range of half-lives with a mean half-live of 11.2 days, with head and stalk subunits displaying more similar half-lives when compared to the whole complex. The nuclear pore complex (NPC) is a huge membrane-spanning protein complex that comprises static scaffold regions, intermediate adopter regions as well as highly dynamic regions (*Rabut et al., 2004*). For analysis of the nuclear pore complex, we used a recent cryo-EM structure of the eukaryotic NPC (*Lin et al., 2016*), which contains 19 distinct proteins. We detected 10 of these proteins and found that the observed half-lives correlate very well with previous observations of subunit stability (*Figure 5D,F*; *Beck and Hurt, 2017*). For example, the large structured solenoid elements of the core and the CNC rings display similar half-lives and they are not predicted to exchange once the complex has been assembled. In contrast, the protein Nup62 displayed the shortest half-life of the observed NPC subunits. Nup62, as part of the Nup54/58/62 channel trimer at the center of the core, is known to dissociate easily from the complex (*Lin et al., 2016*). We also observed a significantly shorter half-life of Sec13, a component of the NPC. However, Sec13 is also a stoichiometric member of the COPII coat so we cannot attribute the observed half-life with dynamic exchange in the NPC, as our data is averaged over all populations in the cell (see Discussion). Others

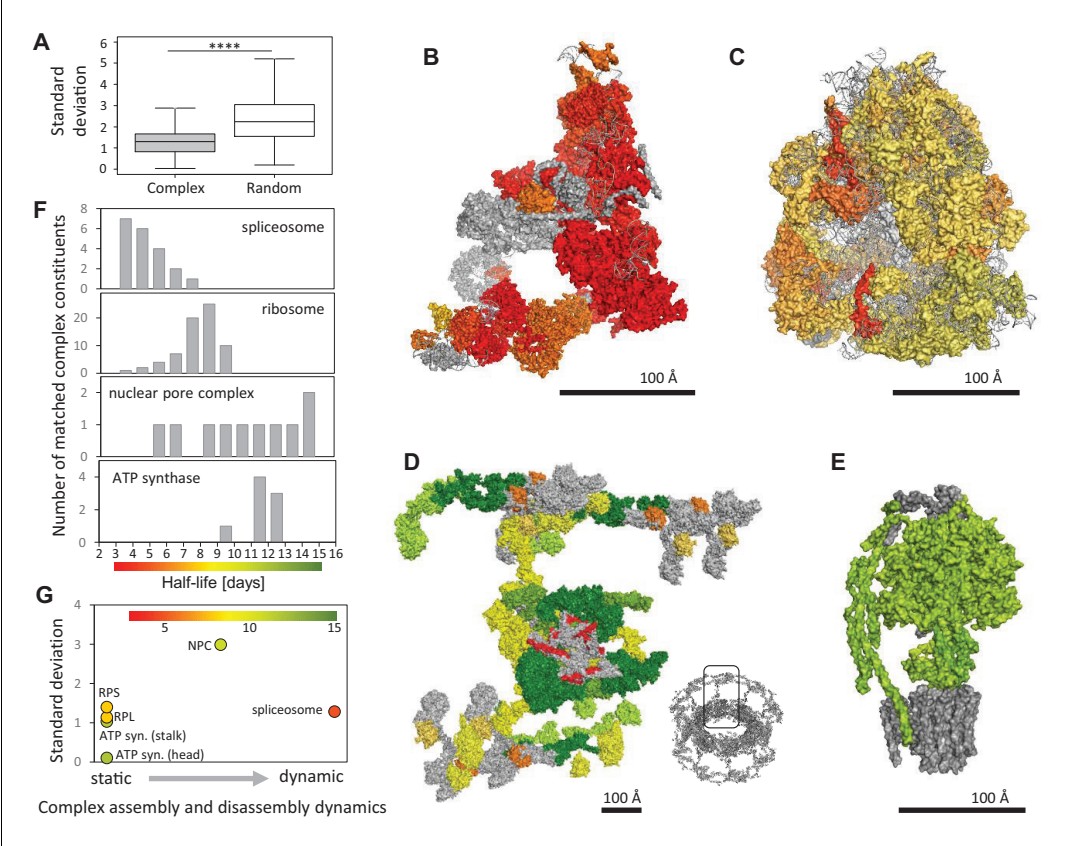

**Figure 5.** Protein half-lives within multi-protein complexes. (**A**) Standard deviation (SD) for half-lives within multi-protein complexes compared to SD between randomly sampled proteins. (**B**) Structures of the spliceosome (PDB code: 5O9Z), (**C**) the ribosome (PDB code: 3J7R), (**D**) the nuclear pore complex (multiple PDB codes, see Materials and methods) and (**E**) the ATP-synthase (PDB code: 5LQX) are color-coded according to the protein half-lives. For the nuclear pore complex (**D**) only one repeating unit is displayed. The whole complex is shown as an insert (bottom right). RNA molecules and proteins for which no half-life was determined are shown in grey. (**F**) Half-life distributions of the complex constituents and the corresponding color scales. (**G**) Relationship between the SD of complex members and complex assembly and disassembly dynamics. Average half-lives of complex members are indicated by the indicated color scale.

DOI: https://doi.org/10.7554/eLife.34202.015

The following source data and figure supplement are available for figure 5:

**Source data 1.** Half-lives of protein complex constituents.
DOI: https://doi.org/10.7554/eLife.34202.017
**Source data 2.** Half-lives of constituents of the ribosome, ATP-synthase, spliceosome and nuclear pore complex.
DOI: https://doi.org/10.7554/eLife.34202.018
**Figure supplement 1.** Correlation of half-life variability within protein complexes and complex size.
DOI: https://doi.org/10.7554/eLife.34202.016

have described some extremely long-lived proteins associated with the NPC in neurons (*in-vivo* study in rat brain; (*Savas et al., 2012*; *Toyama et al., 2013*); we also detect some of the same long-lived NPC components in our neural samples. Of course, we cannot rule out the possibility that all of the complex member proteins that we analyzed were indeed physically associated with their respective complexes. With this caveat in mind, we considered the cohesiveness of the half-lives of the four above protein complexes in a conceptual framework that considers the relative dynamism of the complex assembly and disassembly (*Figure 5G*). At the dynamic extreme is the spliceosome; here were observed that the spliceosome constituents exhibited the fastest turnover of the four complexes examined. At the static extreme, we placed the ATP synthase and the small and the large subunit of the ribosome. Notably, for the ATP synthase, we observed the longest average half-life. The NPC is located in the center of the dynamism axis, as it contains both a static core as wells as dynamic outer filaments. Correspondingly, we found the highest variation of subunit half-lives in the

NPC. However, both the dynamic spliceosome and the static ribosome sub-complexes displayed similar subunit half-life variations.

## Nature vs. nurture in protein turnover

In previous studies, different protein lifetimes have been observed in different cell types (*Mathieson et al., 2018*) and in different tissues (*Price et al., 2010*). We thus asked if the protein lifetimes we examine here are intrinsic to their amino acid sequence (nature) or rather whether there exist environmental influences (nurture), either intracellular (e.g. the specific cell type in which the protein is expressed) or extracellular (the types of cells in the extracellular neighborhood). Indeed, protein degradation and protein stability have been linked to intrinsic sequence properties, such as the N-terminal amino acid, denoted as 'N-end rule' (*Bachmair et al., 1986*; *Gibbs et al., 2014*). Accordingly, we analyzed the N-terminal amino acid sequence for all protein groups that were assigned to one specific protein sequence. Sequence analysis revealed no significantly over- or under-represented N-terminal amino acids in the fraction of short-lived proteins (10% of proteins with shortest half-lives) or long-lived proteins (10% of proteins with longest half-lives), respectively (*Figure 3—figure supplement 1A,B*). Interestingly, we found more proteins with a destabilizing N-terminal amino acid in the fraction of long-lived proteins compared to short-lived proteins (*Figure 3—figure supplement 1C*). In primary hippocampal cultures, N-terminal sequence properties seem to have little impact on protein lifetimes when compared to protein functions and localization.

To examine potential environmental influences, we prepared 'glia-enriched' and 'neuron-enriched' cell cultures. The proteomes of these neuron-enriched and glia-enriched cultures were clearly different from the mixed cultures (*Figure 6—figure supplement 1A,H*) and the respective markers of the inappropriate cell type were significantly de-enriched (*Figure 6—figure supplement 1B,C*, *Figure 6—source data 1*; see Materials and methods). Proteomic analysis further revealed that the glia-enriched cultures are strongly enriched in astrocytes and microglia, while markers for oligodendrocytes do not show a consistent enrichment (*Figure 6—figure supplement 1D,E*; see also *Figure 6—figure supplement 4*).

We next compared protein half-lives in the three different culture types (glial-enriched, neuron-enriched cultures, and standard 'mixed' cultures used for all of the above described experiments; *Figure 6A*). We found that, at the population level, the proteins extracted from the glial-enriched cultures exhibited a significantly faster turnover than the proteins extracted from either the neuron-enriched or mixed cultures, which were much more similar to one another (*Figure 6B*). This global difference in turnover between the proteins from the glial-enriched samples and the other two samples cannot be explained by simple differences in their respective proteomes, as there was an 75% and 78% overlap in proteomes between the glial-enriched proteome and neuron-enriched and mixed proteomes, respectively (*Figure 6—figure supplement 1A*). We next compared the turnover rates of individual proteins shared between any two proteomes. A comparison of the turnover rates in glial-enriched cultures vs mixed cultures revealed a systematic shift of almost the entire proteome to faster half-lives in glial-enriched cultures (*Figure 6C*). We considered whether the faster turnover observed in glial-enriched cultures could be related to ongoing cell-division of glia cells, but, as noted above, there is no significant cell division of glia in the confluent mature cultures (*Figure 6—figure supplement 5*). We further explored the issue of cell division by examining the half-life of Histone H3.1 in the mixed vs. glia-enriched cultures. The heavy isotopically labelled form of Histone H3.1 is only produced by cell division and not by protein turnover in post-mitotic cells (*Ahmad and Henikoff, 2002*; *Schwartz and Ahmad, 2005*; *Wu et al., 1982*). Similar long half-lives for Histone H3.1 were obtained in mixed cultures (11.6 days) and glia-enriched cultures (11.2 days) demonstrating that a similar low rate of cell division takes place in both culture types and that cell division cannot account for the faster protein turnover in glia cells (*Figure 6—figure supplement 5B*).

A comparison of the turnover rates of proteins identified in both the neuron-enriched proteome and the mixed-culture proteome revealed 169 proteins that exhibited a significantly faster turnover in mixed cultures as well as 68 proteins that exhibited a significantly faster turnover in neuron-enriched cultures (*Figure 6D*, *Figure 6—source data 2*). We asked whether any particular functional groups are significantly over-represented using Gene Ontology (*Figure 6—source data 3*). We found that proteins with faster turnover rates in mixed compared to neuron-enriched cultures were significantly over-represented in functional groups that represent ribosomal proteins, cytoplasmic translation, the extracellular matrix and focal adhesion (*Figure 6E*). Further, we found that proteins

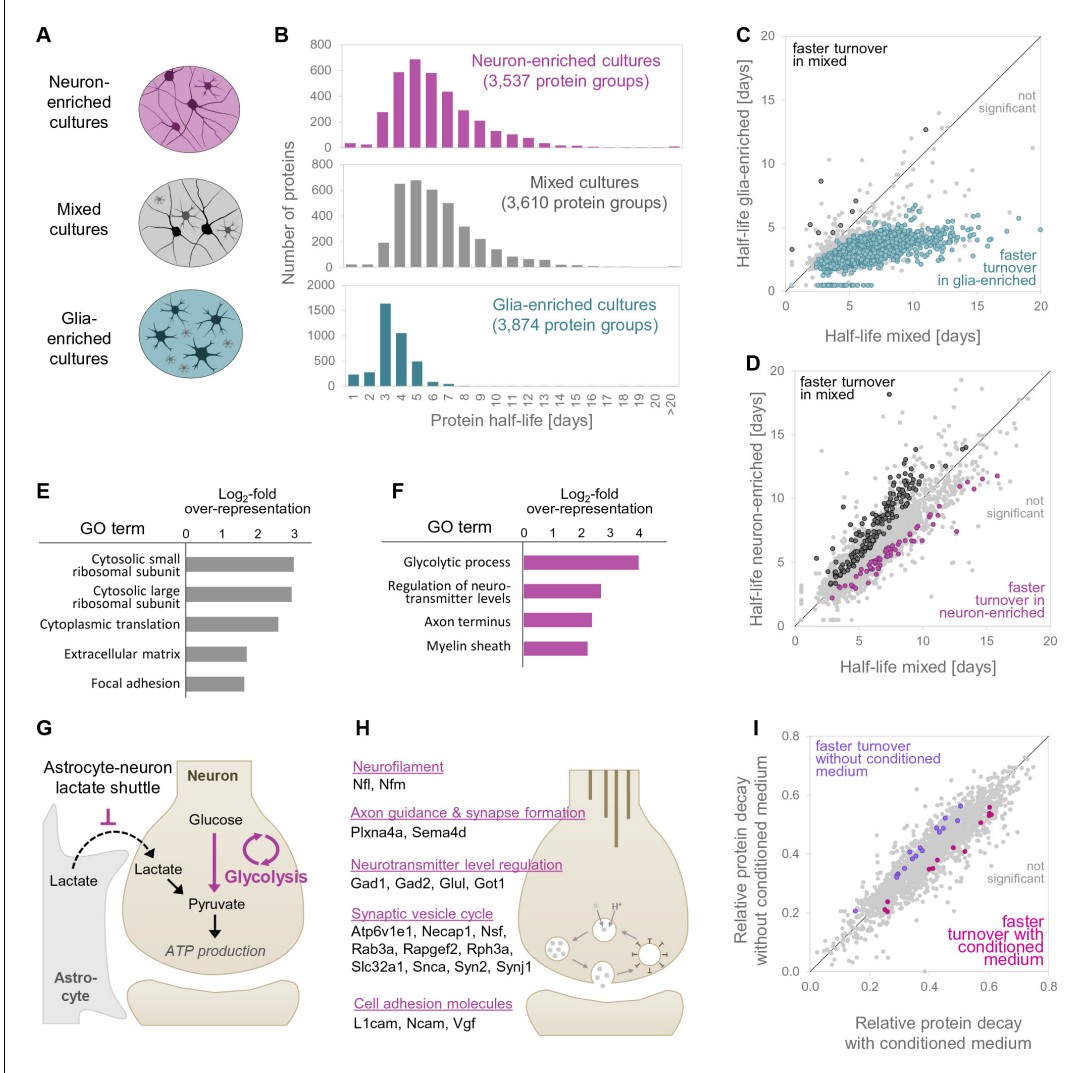

**Figure 6.** Protein turnover rates of different cell types in different cellular environments. (A) Different cell culture types and environments for which protein turnover was investigated. (B) Protein half-life distributions for different primary neuronal culture types (mixed, neuron-enriched and glia-enriched cultures). (C and D) show protein-wise half-life comparisons for proteins in glia-enriched versus mixed cultures as well as neuron-enriched versus mixed cultures. Proteins with significantly regulated turnover rates (p<0.05 at≥2 of 3 time points, Bonferroni corrected) are highlighted in color. (E) GO over-representation analysis of proteins with significantly faster turnover rates in mixed cultures compared to neuron-enriched cultures. Selected over-represented GO terms (p<0.05) and their log$_2$-fold over-representation are shown. (F) Same as E for proteins with significantly faster turnover rates in neuron-enriched cultures compared to mixed cultures. (G) Proposed mechanism to explain the increased turnover of glycolytic enzymes in neuron-enriched cultures. In the absence of glia cells, neurons are not supplied with lactate via the astrocyte-neuron-lactate-shuttle and hence pyruvate production relies on glycolysis exclusively. Glycolytic enzymes might be used more frequently, damaged more frequently and replaced earlier. (H) Selected proteins with significantly faster turnover rates in neuron-enriched cultures that are involved in synaptic processes. (I) Protein-wise turnover comparisons for proteins from neuron-enriched cultures maintained with or without conditioned medium. Proteins with significantly regulated turnover rates (p<0.05, Bonferroni corrected) are highlighted in color.

DOI: https://doi.org/10.7554/eLife.34202.019

The following source data and figure supplements are available for figure 6:

**Source data 1.** Relative protein abundances in different culture types.
DOI: https://doi.org/10.7554/eLife.34202.025
**Source data 2.** Protein half-life comparison between different culture types.
DOI: https://doi.org/10.7554/eLife.34202.026
**Source data 3.** GO analysis of proteins with different turnover rates in mixed and neuron-enriched cultures.
DOI: https://doi.org/10.7554/eLife.34202.027
**Source data 4.** Proteins with different turnover rates in neuron-enriched cultures with and without conditioned medium.

*Figure 6 continued on next page*

*Figure 6 continued*

DOI: https://doi.org/10.7554/eLife.34202.028

**Figure supplement 1.** Comparison of different culture types.

DOI: https://doi.org/10.7554/eLife.34202.020

**Figure supplement 2.** Faster protein turnover of glycolytic enzymes in neuron-enriched cultures.

DOI: https://doi.org/10.7554/eLife.34202.021

**Figure supplement 3.** Comparison of neuron-enriched cultures with and without conditioned medium.

DOI: https://doi.org/10.7554/eLife.34202.022

**Figure supplement 4.** Proteomic comparison of glia-enriched cultures prepared from cortex and hippocampus.

DOI: https://doi.org/10.7554/eLife.34202.023

**Figure supplement 5.** Estimation of cell division in primary cultures.

DOI: https://doi.org/10.7554/eLife.34202.024

with faster turnover rates in neuron-enriched cultures were significantly over-represented in functional groups related to glycolytic processes, the regulation of neurotransmitter levels, axon terminus and myelin sheath (*Figure 6F*). Within the glycolysis pathway, we discovered that a majority of the enzymes exhibited faster turnover in neuron-enriched cultures (*Figure 6—figure supplement 2*). It has been demonstrated that astrocytes provide lactate (which can be converted to pyruvate by the lactate dehydrogenase) to neurons (*Magistretti and Pellerin, 1999*; *Mason, 2017*); the relative paucity of glia in the neuron-enriched cultures would lead to the loss of astrocyte-derived lactate and could lead to enhanced metabolic activity of the neuron's own glycolytic axis, and a resulting faster turnover (*Figure 6G*). Within the synapse, we noted that several proteins related to the synaptic vesicle cycle as well as neurotransmitter synthetic enzymes exhibited a faster turnover in neuron-enriched cultures relative to mixed cultures (*Figure 6H*).

In the above experiments, differences in protein turnover in the neuron-enriched versus mixed (neuron-glia) cultures could be due to secreted factors or due to the physical presence of glial cells. To address this, we asked whether the composition of the extracellular medium alone could influence protein turnover. We compared the protein turnover rates in 'sister' neuron-enriched cultures grown in a medium supplemented with conditioned medium (obtained from glia cells and mixed cortical cultures) or without conditioned medium. As above, we used SILAC to assess protein turnover, performing the medium swap (light-to-heavy or heavy-to-light; see Materials and methods) at DIV 18–19. We observed that cell density was clearly lower for neuron-enriched cultures without conditioned medium compared to sister-cultures that were supplemented with conditioned medium although cell viability was similar (visual inspection). When we compared the protein turnover rates in the two conditions, we found that they were very similar to one another, with just a few proteins showing altered turnover as a result of the presence or absence of pre-conditioned medium (*Figure 6I*). We also noted that the proteins that had a faster turnover in neuron-enriched cultures compared to mixed cultures (see *Figure 6D*) did not show different turnover kinetics in neuron-enriched cultures with and without conditioned medium (e.g. glycolytic enzymes and proteins depicted in *Figure 6H*; see also *Figure 6—source data 4*). These data indicate that the differences in protein turnover we observe between neuron-enriched and neuron-glia cultures likely arise from physical/mechanical interactions between the cells, presumably owing to adhesive interactions and its associated signaling.

## Discussion

Here, we used a dynamic SILAC approach to describe the turnover rates of neuronal proteins in different cellular environments. All in all, we obtained half-life measurements for over 5100 protein groups which exhibited a range of half-lives from <1 day to >20 days, with a median half-life of 5.4 days in mixed cultures. A similar half-life distribution (median = 4.2 days) was obtained by *Cohen et al. (2013)* who measured the turnover of 2802 protein groups from mixed cortical cultures. A recent study by *Mathieson et al. (2018)* reported clearly shorter half-lives for primary mouse embryonic neurons (median = 1.9 days). These shorter half-lives are likely explained by the relative immaturity of the cultures (prepared from embryonic rather than postnatal neurons) and the relative short time *in vitro* (5 days vs. 18–19 days). We further compared the protein half-lives

determined by our *in vitro* study in mixed cultures to half-lives of mouse brain proteins that were previously determined by *in vivo* metabolic labelling. Though proteins were clearly longer-lived *in vivo* (average half-life ~9 days) compared to *in vitro*, a gene-wise comparison revealed a significant correlation between protein half-lives (Spearman rank correlation = 0.62, p<0.001; *Figure 1—figure supplement 3*). The systematic offset could be explained by the different experimental approach, species (mouse versus rat), cellular compositions and environments as well as by the different ages of the investigated cells. *Visscher et al. (2016)* recently reported shorter half-lives for proteins in young cells (*C. elegans* larvae) compared to older cells (adult *C. elegans*).

We validated the protein turnover rates we measured using dynamic SILAC with a metabolic labelling and visualization strategy, FUNCAT-PLA (*tom Dieck et al., 2015*). We found that the ranked half-lives of the proteins determined via MS and FUNCAT-PLA were very well correlated. The half-lives of proteins visualized *in situ* with FUNCAT-PLA, however, were systematically shorter than those determined via MS. Reasons for this offset could include a relative instability of the azide group or AHA-containing proteins, or an increased sensitivity of the AHA-labeling to detect the rapid degradation of misfolded proteins which are missed by the continuous SILAC labelling (also discussed below).

Analysis of our dynamic SILAC data revealed that protein turnover is influenced by the cellular location of the protein. We found that membrane proteins, in particular those of the plasma membrane, the ER, and the Golgi apparatus exhibited significantly shorter than average half-lives while mitochondrial proteins exhibited significantly longer half-lives. These systematic differences in protein turnover might result from different major degradation mechanisms for cytosolic and membrane proteins (*Jin et al., 2018*; *Tai and Schuman, 2008*) as well as the unique quality control mechanism of mitochondria by fission and fusion (*Cagalinec et al., 2013*; *Gomes and Scorrano, 2008*). Interestingly, the population of synaptic proteins exhibited a similar distribution of half-lives to the general population. Protein half-lives also correlate with protein functions. Receptors, signalling molecules and proteins involved in cellular communication, for instance, were over-represented in the group of short-lived proteins. In contrast, other functional classes such as histones and proteins involved in energy metabolism were over-represented in the fraction of long-lived proteins. This finding supports the idea that protein lifetimes evolved evolutionary in a way that proteins involved in internal and external signal response are short-lived to enable faster and more fine-tuned regulation, while proteins that serve more steady functions in the cell are longer-lived in order to save energy that is required for protein synthesis and degradation.

Proteins are interactive by design and this interaction is essential for the execution and regulation of cellular functions. *Cohen et al. (2013)* showed that interacting proteins have more similar half-lives compared to non-interacting proteins in primary cortical neurons. Expanding on this, we found that members of multi-protein complexes display more similar half-lives than randomly selected proteins. Protein complexes differ greatly in their biogenesis, assembly and disassembly kinetics, and these differences might be reflected in different mean half-lives between complexes or variability of half-lives within complexes. We examined in detail the half-lives of protein constituents of 4 different multi-protein complexes: the ATP synthase, the ribosome, the nuclear pore complex and the spliceosome. For the four complexes we examined, the most dynamic in terms of assembly and disassembly is the spliceosome and the proteins in this complex exhibited the shortest half-lives. The nuclear pore complex exhibited highly variable half-lives of its associated protein constituents, which is in agreement with previous studies (*Beck and Hurt, 2017*; *Daigle et al., 2001*; *Mathieson et al., 2018*). Surprisingly, the ribosome, a canonical stable cellular complex where the 40S and 60S subunits are thought to be assembled 'for life' in the nucleus and nucleolus, exhibited substantial variability in the half-lives of its constituents, suggesting the possibility that some ribosomal proteins may be exchanged to repair (*Pulk et al., 2010*) or even specialize ribosomes (*Xue and Barna, 2012*) in subcellular compartments (*Holt and Schuman, 2013*).

The above results must be interpreted with the caveat that we do not know if the half-lives we measured represent a single functional state for the proteins. A recent study in mouse fibroblasts demonstrated that ~ 10% of the proteins show non-exponential decay kinetics with a faster degradation in the first hours after synthesis (*McShane et al., 2016*). The dynamic SILAC approach we used here does not allow us to accurately distinguish between different sub-populations of the same protein that have different stabilities. As unstable proteins are degraded shortly after synthesis, they represent a low abundant sub-population in the whole cell lysate and contribute less to the

measured MS signals compared to the corresponding stable protein population. Furthermore, we did not acquire samples in the first hours after the medium change, which would be necessary to accurately resolve protein populations with different decay kinetics. In contrast, sub-populations with different lifetimes could be better detected in a pulse-chase experiment sampling multiple chase time points in short time intervals after the pulse. During the pulse, proteins of all sub-populations are synthesized and during the chase these sub-populations will decay with different time constants: the fraction of short-lived proteins will vanish after a short time while the longer-lived fraction will remain much longer before it eventually degrades. Alternatively, in future experiments, the purification of subcellular compartments (e.g. synaptosomes) or organelles (e.g. ribosomes) and an independent determination of the half-lives of proteins in particular cell locations or functional groups could provide new insights on sub-populations with different half-lives.

Factors intrinsic to the amino acid sequence have been described, which contribute to the stability of proteins; specific amino acids were found to have a destabilizing effect when located at the protein N-terminus (*Bachmair et al., 1986*; *Gibbs et al., 2014*). In our dataset, there was no over-representation of destabilizing N-terminal amino acids in the fraction of short-lived proteins compared to long-lived proteins, indicating that N-terminal sequence properties have little impact on protein lifetimes in primary hippocampal cultures.

We further examined whether there is an additional effect of the intracellular environment on brain protein turnover by comparing the half-lives of the same proteins in either glial- or neuron-enriched cultures. We found that when expressed in glia, proteins exhibited a significantly faster turnover rate than they did when expressed in neurons; this difference cannot be explained by glial cell division. Others have investigated turnover rates for nuclear proteins in different brain cell types and also observed faster protein turnover for glia cells compared to neurons (*Toyama et al., 2013*). We also examined the effect of the extracellular environment on protein turnover, comparing the turnover of proteins in neuron-enriched cultures to mixed cultures, which contain both neurons and glia. We found that while the majority of protein half-lives were not changed between the two conditions, a significant group of proteins exhibited faster or slower turnover when glia were present in the extracellular environment. We believe that most of the changes in half-life are likely due to adhesive interactions with glial cells, rather than secreted factors, as there were no big differences in turnover when we compared neuron-enriched cultures treated, or not, with glial conditioned medium. Proteins that showed faster turnover in the absence of glia included both glycolytic enzymes and proteins associated with vesicle cycling. This turnover regulation is likely related to the activity state of the respective proteins. In the absence of glia cells, neurons lack lactate supplied from glia (by the glia-neuron lactate shuttle), and thus require increased glycolytic activity to produce pyruvate. The observed increase in turnover of glycolytic enzymes might reflect this increased activity. A similar correlation between turnover and the activity of individual proteins was recently demonstrated in yeast (*Martin-Perez and Villén, 2017*). Taken together, these data indicate a potent influence of both the intracellular (local) and extracellular (global) environment on protein turnover.

In the present experiments we rely on the relative enrichment of neurons or glia in special culture conditions. With the recent development of cell type-specific metabolic labelling (*Alvarez-Castelao et al., 2017*), there exists the possibility to directly examine the protein turnover of distinct cell types in combination with the dynamic SILAC approach employed here. In addition, understanding how plasticity regulates and exploits turnover to modify brain proteomes is a goal of future studies.

## Materials and methods

### Preparation and maintenance of cultured neurons and glia cells

Mixed cultures

Dissociated hippocampal neurons were prepared and maintained as previously described (*Aakalu et al., 2001*). Briefly, hippocampi from postnatal day 1 rat pups (strain Sprague-Dawley) were dissected and dissociated by papain and plated at a density of 18,000 cells/cm² onto poly-D-lysine-coated Petri dishes (MatTek, Ashland, Massachusetts). Cultures were maintained in Neurobasal-A medium (Invitrogen, Carlsbad, California) supplemented with B-27 (Invitrogen) and Glutamax

(Invitrogen) at 37°C. All experiments were carried out with the approval of the German animal experiment authorities.

## Primary hippocampal cultures for FUNCAT-PLA experiments

For FUNCAT-PLA experiments, primary hippocampal neurons were prepared as described above and plated at a density of approximately 20,000 cells/cm$^2$ onto poly-D-lysine-coated glass-bottom Petri dishes (MatTek).

## Neuron-enriched cultures

Hippocampal cells were dissociated and plated as described above at a density of approximately 25,000 cells/cm$^2$. After one day, 3 µM cytosine β-D-arabinofuranoside (AraC; Sigma-Aldrich, St. Louis, Missouri) was added for 48 hr. Subsequently, the cells were maintained in a medium composed of Neurobasal-A supplemented with B-27 and Glutamax of which 20% were pre-conditioned over cortical neurons and 30% were pre-conditioned over glia cells.

## Neuron-enriched sister cultures with and without pre-conditioned medium

Hippocampal cells were dissociated and plated as described above and plated at a density of approximately 25,000 cells/cm$^2$. Eight hours after plating, the medium was fully exchanged with 'light' or 'heavy' (K10, R8) Neurobasal-A supplemented with B-27 and Glutamax (NGM) or with 'light' or 'heavy' (K10, R8) Neurobasal-A supplemented with B-27 and Glutamax of which 20% were pre-conditioned over cortical cultures and 30% were pre-conditioned over glia cells (NGM- conditioned). One day after plating, 3 µM AraC was added for 48 hr. Subsequently, the cells were maintained again in 'heavy' or 'light' NGM or NGM conditioned as before.

## Glia-enriched cultures

The cortex from postnatal day 1 rat pups was dissected and dissociated as described above. 20% of the cell suspension was plated out and maintained on non-coated 6 cm Petri dishes (MatTek). Cells were initially grown in MEM++. Medium was exchanged with fresh MEM++ after 4 hr and after 4 days. After 7 days in culture, medium was exchanged by conditioned medium composed of Neurobasal-A supplemented with B-27 and Glutamax of which 5% were pre-conditioned over cortical neurons and 15% were pre-conditioned over glia cells.

## Comparison of glia-enriched cultures prepared from cortex and hippocampus

The cortex from a postnatal day 1 rat pup was dissected and dissociated by papain. Five percent of the cell suspension was plated on non-coated 3 cm Petri dishes (MatTek). Four hippocampi from two rat pups (postnatal day 1) were dissected and dissociated by papain. Twenty-five percent of the pooled cell suspension was plated on non-coated 3 cm Petri dishes. Cells were maintained in MEM++ at 37°C. After 11 days in culture, cells were washed with ice-cold DPBS (Invitrogen) supplemented with protease inhibitor (cOmplete EDTA-free, Roche, Basel, Switzerland), scraped, pelleted by centrifugation, frozen in liquid nitrogen and stored at −80°C.

## Dynamic SILAC experiment in mixed, neuron-enriched and glia-enriched cultures

After 18–19 days in culture, the growth medium was exchanged with a medium that was depleted of arginine and lysine (customized; Invitrogen) and added 'heavy' isotopically labeled arginine (R10; Thermo Fisher, Waltham, Massachusetts) and lysine (K8; Thermo Fisher) resulting in a final percentage of 80% Arg10/Lys8% and 20% remaining 'light' arginine (R0)/lysine (K0). Visual inspection of the cells in pilot experiments revealed improved cell viability when a thin layer of initial medium remained on the cells during the medium change compared to a complete medium removal and addition of fresh medium or medium that was pre-conditioned in sister-cultures. Cells were harvested 0, 1, 3 and 7 days after the medium switch. Cells were washed with ice-cold DPBS supplemented with protease inhibitor (cOmplete EDTA-free, Roche), scraped, pelleted by centrifugation, frozen in liquid nitrogen and stored at −80°C. For each culture type, three independent biological replicates were performed.

## Dynamic SILAC experiment in neuron-enriched sister cultures with and without conditioned medium

After 18–19 days in culture, approximately 80% of the medium was swapped between a heavy and light neuron-enriched cultures without conditioned medium or between heavy and light neuron-enriched cultures with conditioned medium, respectively. Cells were harvested 7 days after the medium switch. For each culture type, one dish was also harvested just before the medium change (0 day). Two independent biological replicates were performed.

## Sample preparation for MS analysis

The cell pellets were lysed in lysis buffer (8 M urea, 200 mM Tris/HCl [pH 8.4], 4% CHAPS, 1 M NaCl, cOmplete EDTA-free protease inhibitor) using a pistil and sonication for 4 × 30 s at 4°C. The samples were incubated with Benzonase (1 µL of a $\geq$ 250 units/mL stock solution; Sigma) for 10 min, and centrifuged for 5 min at 10,000 x g to clear the samples from cell debris.

The proteins were digested as described by *Wiśniewski et al. (2009)*. In brief, the samples were mixed with Urea Buffer A (UA: 8 M urea, 0.1 M Tris/HCl [pH 8.5]), loaded onto a 10 kDa Microcon filter (Merck Millipore, Burlington, Massachusetts) and centrifuged at 17,000 x g for 20 min. After an additional wash step with UA, the proteins on the filter membrane were reduced with 10 mM TCEP (in UA) for 30 min. TCEP was subsequently removed by centrifugation and proteins were then added 95 mM IAA in UA and incubated in the dark for 30 min. IAA was removed by centrifugation and the samples were washed three times with 100 µL of Urea Buffer B (UB: 8 M urea, 0.1 M Tris/HCl [pH 8.0]). Proteins were digested with Endopeptidase LysC (Promega, Madison, Wisconsin) in a protein to enzyme ratio of 50:1 and subsequently with trypsin (Promega) in a protein to enzyme ratio of 100:1. Samples were desalted using C18 StageTips (*Rappsilber et al., 2007*) or SepPak cartridges (Waters, Milford, Massachusetts) as described by *Schanzenbächer et al. (2016)*. Samples were dried by vacuum centrifugation and stored at −20°C until LC-MS analysis.

## LC-MS/MS Analysis

The dried peptide samples were reconstituted in 5% acetonitrile with 0.1% formic acid and subsequently loaded using a nano-HPLC (Dionex U3000 RSLCnano) onto a PepMap100 loading column (C18, L = 20 mm, 3 µm particle size, Dionex, Sunnyvale, California) and washed with loading buffer (2% acetonitrile, 0.05% trifluoroacetic acid in water) for 6 min at a flow rate of 6 µL/min. Peptides were separated on a PepMap RSLC analytical column (C18, L = 50 cm, <2 µm particle size, Dionex) by a gradient of phase A (water with 5% v/v dimethylsulfoxide and 0.1% formic acid) and phase B (5% dimethylsulfoxide, 15% water and 80% acetonitrile v/v/v). The gradient was ramped from 4% B to 48% B in 178 min at a flow rate of 300 nL/min. All solvents were LC-MS grade and purchased from Fluka. Peptides eluting from the column were ionized online using a Nanospray Flex ion source (Thermo Scientific) and analyzed either in a 'Q Exactive Plus' (Thermo Scientific) or in an 'Orbitrap Elite' (Thermo Scientific) mass spectrometer in data-dependent acquisition mode. For Q Exactive Plus measurements, precursor ion spectra were acquired over the mass range 350–1400 m/z (mass resolution 70 k), and the top10 precursor ions were selected for fragmentation (HCD; normalized collision energy = 30) and analysis in MS2 mode (resolution 17.5 k). For Orbitrap Elite measurements, precursor ion spectra were acquired over the mass range 350–1600 m/z (FTMS; mass resolution 120 k), and the top15 precursor ions were selected for fragmentation (CID; normalized collision energy = 35) and analysis in MS2 mode (ITMS). The full parameter sets are listed in *Supplementary file 2*. All dynamic SILAC samples were measured in triplicate LC-MS/MS runs. The samples for comparison of glia-enriched cultures obtained from cortex and hippocampus were measured in technical duplicates.

## Database searches

Raw data were analyzed with MaxQuant (version 1.6.0.1; RRID:SCR_014485 [*Cox and Mann, 2008*; *Tyanova et al., 2016a*]) using customized Andromeda parameters (see *Supplementary file 3*).

For all searches, spectra were matched to a *Rattus norvegicus* database downloaded from uniprot.org (37,669 entries, reviewed and unreviewed; RRID:SCR_002380) and a contaminant and decoy database. Precursor mass tolerance was set to 4.5 ppm, fragment ion tolerance to 20 ppm (QExactive Plus) or 0.5 Da (Orbitrap Elite), respectively. Carbamidomethylation (+57.021) of cysteine

residues was set as fixed modification and protein-N-terminal acetylation (+42.011) as well as methionine oxidation (+15.995) were set as variable modifications. A False discovery rate of 0.01 was applied at the PSM and protein level. If not stated otherwise, only unique peptides were included in down-stream analysis. For the dynamic SILAC samples, two different parameter sets were used to (a) relatively quantify protein abundance and assess the rate of incorporation of 'light' amino acids into nascent proteins during the 'heavy' pulse (R10 and K8 as variable modifications) and to (b) quantify 'heavy/light' ratios (R10 and K8 as heavy SILAC labels). For (a), heavy arginine (R10; +10.008) and heavy lysine (K8; +8.014) were set as additional variable modifications, whereas for (b), R10 and K8 were set as heavy SILAC partners (multiplicity = 2).

All proteomics data associated with this manuscript have been uploaded to the PRIDE online repository (*Vizcaíno et al., 2013*).

## Bioinformatic processing and data analysis

### Relative protein quantification in dynamic SILAC samples

Protein results from MaxQuant search B were filtered to remove decoys and contaminants and protein intensities were subsequently normalized to the mean intensity of each injection (*Figure 6— source data 1*). GO over-representation analysis of neuron-related proteins (detected in ≥2 neuron-enriched samples and absent in all glia-enriched samples) and glia-related proteins (detected in ≥2 glia-enriched samples and absent in all neuron-enriched samples), respectively, was performed using the Gene List Analysis tool of the Panther Classification System (RRID:SCR_015893; *Mi et al., 2013*). All identified proteins were used as reference data set. Significantly over-represented GO terms (p<0.05; Bonferroni corrected) are shown in *Figure 6—figure supplement 1D,E*. For principle component analysis (*Figure 6—figure supplement 1F*; performed using Perseus software package; version 1.5.2.6; RRID:SCR_015753 (*Tyanova et al., 2016b*), protein intensities were averaged within biological replicates and $\log_2$ transformed. Only proteins with expression values for all biological replicates could be used for principle component analysis.

For a quantitative comparison of individual proteins, protein intensities of technical replicates were averaged. To characterize the cellular composition of the different culture types, relative expression levels of established neuronal and glial marker proteins as well as previously reported astrocyte, oligodendrocyte and microglia marker proteins (the top 20 most enriched proteins in the respective cell types reported by *Sharma et al., 2015*) were compared between the different culture types (*Figure 6—figure supplement 1B–E*). Forty-three of these astrocyte, oligodendrocyte and microglia marker proteins, that were detected in mixed cultures at all time points within a samples set (biological replicate), were used to quantify the relative expression levels of glia proteins over the time course of the dynamic SILAC experiment (7 days; see *Figure 6—figure supplement 5A*).

### Incorporation of light arginine and lysine into nascent proteins

Peptide results from MaxQuant search A were filtered for decoys and contaminants. Peptides containing two arginine and/or lysine residues (due to a missed tryptic cleavage site) were extracted and the ratio of the following combinations was calculated for each sample (except $t_0$ samples) based on the number of detections: 'light-heavy' and 'heavy-heavy'. The probabilities of incorporation of a light or heavy amino acid, respectively, into a nascent protein was calculated using the following equation system:

$$P(L) + P(H) = 1 \tag{1}$$

$$2 \cdot P(L) \cdot P(H) = \frac{LH}{LL + LH + HH} \tag{2}$$

$$P(H) \cdot P(H) = \frac{HH}{LL + LH + HH} \tag{3}$$

LL: Newly synthesized peptides containing two 'light' Arg/Lys; cannot be experimentally assessed, since newly synthesized 'light' peptides cannot be distinguished from pre-existing peptides

LH: Newly synthesized peptides containing one 'light' and one 'heavy' Arg/Lys; experimentally assessed for each sample

HH: Fraction of newly synthesized peptides containing two 'heavy' Arg/Lys; experimentally assessed for each sample

P(L): Probability of incorporation of a 'light' Arg/Lys into a nascent protein

P(H): Probability of incorporation of a 'heavy' Arg/Lys into a nascent protein

Combination of *equations 1-3* lead to:

$$P(H) = \frac{2 \cdot \frac{HH}{LH}}{1 + 2 \cdot \frac{HH}{LH}} \tag{4}$$

$$P(L) = 1 - P(H) \tag{5}$$

For protein half-life determination, P(H) is used as a correction factor to convert the fraction of 'light' peptides into the fraction of pre-existing peptides (see below).

## Protein half-life determination

Peptide results from MaxQuant search B were filtered for decoys and contaminants. The fractions of remaining light peptides (%L) were calculated for each measurement and each peptide based on the H/L ratios (computed by MaxQuant) using *equation 6*. For $t_0$ samples, %L was set to 1, if the peptide was only detected in its 'light' form and hence no H/L was computed.

$$\%L = \frac{1}{1 + H/L} \tag{6}$$

%L: fraction of remaining 'light' peptide

H/L: Heavy-light ratio computed by MaxQuant

The fraction of light peptide was converted into the fraction of pre-existing peptide (%old) using a correction factor that corrects for incorporation of light Arg/Lys into newly synthesized proteins, see *equation 7*. Resulting negative values were excluded from further analysis.

$$\%old = 1 - \frac{1 - \%L}{P(H)^{MC}} \tag{7}$$

%old: fraction of pre-existing peptide

P(H): Correction factor; probability of incorporation of a 'heavy' Arg/Lys into a nascent protein

MC: number of missed cleavages

Peptides were subsequently filtered within biological replicates. Only peptides that were quantified at all four time points ($t_0$, 1d, 3d, 7d) and with a mean %old >0.9 at $t_0$ were considered for further analysis. Only peptides unique for a protein group were used for further analysis. In a few cases, two protein groups were merged (in order to rescue peptides) and only peptides unique for the merged group were further considered. For protein group 5450 (Rps27A) peptides assigned to the sequence of ubiquitin were removed, so that all remaining peptides were specific for Rps27A. The filtered peptide data of the biological replicates was merged. For each culture type, protein group and time point, %old values that were identified as outliers (%old <quartile 25–1.5*interquartile range or %old >quartile 75 + 1.5*interquartile range) were removed. Protein groups which average %old did not show a continuous decay over time were excluded from further analysis. Hierarchical clustering (kmeans method; based on average %old values per protein group) was applied to find groups of proteins with similar turnover kinetics (*Figure 1—figure supplement 4*). Cluster A and B (*Figure 1—figure supplement 4*) contain proteins with comparably short life-times ($\leq$50% remaining pre-existing protein after 1 day) that cannot be accurately fit and were hence assigned a half-life of '<1 day'. For all other proteins, the peptide data (%old over time) were ln transformed and fitted by a linear function, as described in *equation 8*. Protein half-lives ($t_{1/2}$) were calculated based on the rate constant k (negative value of slope of the fit), see *equation 9*. Half-lives, rate constants, standard errors of the slope and coefficients of determination ($R^2$) are given for all proteins and all culture types in *Figure 1—figure supplement 2* and *Supplementary file 1*.

$$\ln(\%old) = -k \cdot t \tag{8}$$

k: rate constant of protein turnover t: Incubation time with "heavy" medium

$$t_{1/2} = \frac{\ln(2)}{k} \tag{9}$$

## Analysis of protein half-lives in mixed cultures

### N-end rule

N-terminal amino acid sequences (excluding initiator methionine) were analyzed for 1797 protein groups that contain only one protein. Sequence over-representation analysis was performed using pLogo (*O'Shea et al., 2013*).

### Subcellular localization

Subcellular localizations were assigned to protein groups from mixed cultures using the LocTree3 database (*Goldberg et al., 2014*). Only proteins assigned to a single localization with a score >50 were considered for analysis (*Figure 3A*; *Figure 3—source data 1*). Synaptic proteins were extracted based on GO term annotations (Uniprot) and literature (*Figure 3A*, *Figure 4*, and *Figure 4—source data 1*). Half-lives at different cellular localizations were compared by a Mann-Whitney test (Bonferroni correction for multiple testing).

### Functional analysis

Proteins from mixed cultures were divided into eight overlapping half-life bins. GO over-representation analysis was performed for proteins in each half-life bin using the Gene List Analysis tool of the Panther Classification System (*Mi et al., 2013*). All protein half-lives were used as reference data set. Significantly over-represented GO terms (p<0.05; Bonferroni corrected) are shown in *Figure 3B,C* and *Figure 3—source data 2*).

### Analysis of protein complexes

Half-lives of proteins that belong to multi-protein complexes were extracted from the mixed culture dataset and standard deviations (SD) between the half-lives of proteins belonging to the same complex were calculated. Protein complex information were obtained from the CORUM database (*Ruepp et al. (2010)*; RRID:SCR_002254) and only complexes composed of ≥5 complex members, for which ≥ 3 half-lives were assigned, were used for analysis (*Figure 5—source data 1*). As comparison, the SD was calculated between random half-lives sampled from the mixed culture dataset (same number of groups and same group sizes as the complexes). Distribution of complex SD and random SD was compared by Mann-Whitney test.

For more detailed analysis of selected complexes, the protein half-lives of the complex constituents were plotted onto the known structures of the spliceosome from *Homo sapiens* (pdb code 5O9Z), the ribosome from *Sus Scrofa* (pdb code 3J7R), the nuclear pore complex from *Saccharomyces cerevisiae* and *Chaetomium thermophilum* (multiple pdbs merged into a full assembly as described by *Lin et al. (2016)*, including pdb codes: 5HAX, 5HAY, 5HAZ, 5HB4, 5HB5, 5HB6, 5HB7, 5HB6, 5HB7 and 5HB8) and the ATP synthase from *Pichia angusta* (pdb code 5LQX); see *Figure 5B–E*. Proteins were matched to homologue chains in the pdb files using gene names when applicable, and half-lives were visualized using a custom red-yellow-green color palette that ranges from 3 to 15 days as indicated in *Figure 5F*. Identical polypeptide chains in different locations in the complex were counted as a single protein for histogram calculation. The protein complexes were plotted in 'surface' representation using PyMOL (RRID:SCR_000305). Subunits for which no half-life was determined are indicated in grey, RNA is plotted in 'cartoon' mode in light grey when applicable.

## Analysis of protein turnover rates in different culture types

All proteins for which half-lives were determined in the mixed and neuron-enriched culture as well as in the mixed and glia-enriched culture were compared protein-wise in order to identify proteins that display different turnover rates in different culture types (*Figure 6B–C*). Proteins which average %old was always greater (or always smaller) at each time point (1d, 3d, 7d) in the cell-type enriched culture

compared to the mixed culture type, were used for statistical analysis. A t-test was performed for each time point to compare %old values between each two different culture types (p<0.05; Bonferroni correction for the total number of tests). A protein's turnover is considered to be significantly different between two cultures types, if %old values are significantly different for at least two of three time points (*Figure 6—source data 2*).

To identify common properties/functions of proteins that showed a significantly faster or slower turnover, respectively, in neuron-enriched cultures versus mixed cultures, we performed a GO over-representation analysis using the Gene List Analysis tool of the Panther Classification System (*Figure 6D–H*; [*Mi et al., 2013*]). As reference data set, all proteins were used for which a turnover rate was determined in both, mixed and neuron-enriched cultures. All significantly over- and under-represented GO terms (p<0.05; Bonferroni corrected) are given in *Figure 6—source data 3*.

## Proteomic comparison of neuron-enriched sister cultures with and without conditioned medium

Protein results from MaxQuant search A were filtered to remove decoys and contaminants. LFQ intensities were averaged across technical replicates and $\log_2$ transformed.

## Comparison of protein turnover in neuron-enriched sister cultures with and without conditioned medium

Peptide results from MaxQuant search B were filtered for decoys and contaminants. The fractions of remaining old peptides were calculated as described above. As a measure for protein turnover, the relative decay of pre-existing peptides was calculated as the difference of pre-existing peptides at time point $t_0$ and 7 days. Peptides with resulting negative values were removed. A pair-wise t-test (peptide level) was applied to compare the relative protein decay in neuron-enriched cultures with and without conditioned medium. Proteins with significantly different turnover (p<0.05; Bonferroni corrected) are reported in *Figure 6I* and *Figure 6—source data 4*.

## Comparison of glia-enriched cultures prepared from cortex and hippocampus

Protein results were filtered to remove decoys and contaminants. Protein intensities and LFQ values were averaged across technical replicates and $\log_2$ transformed. Protein intensities were used to compare the number of quantified proteins in each sample (*Figure 6—figure supplement 4A*) and LFQ intensities were used for a protein-wise comparison of expression levels in the different samples (*Figure 6—figure supplement 4B*).

## FUNCAT-PLA

FUNCAT-PLA experiments were performed as described by *tom Dieck et al. (2015)*. After 18 days in culture, cells were incubated in methionine-free Neurobasal-A (custom made by Invitrogen) supplemented with 4 mM azidohomoalanine (AHA; in-house synthesized) for 2 hr. In methionine control samples, the medium was supplemented with 4 mM methionine (Sigma) instead of AHA. After metabolic labeling, cells were placed back into their original medium for different chase times (1–7 days as indicated). Subsequently, cells were washed with PBS-MC (1 × PBS, pH 7.4, 1 mM $MgCl_2$, 0.1 mM $CaCl_2$; Gibco), fixed in PFA-sucrose (4% paraformaldehyde (Alfa Aesar), 4% sucrose in PBS-MC) and stored in PBS at 4°C until further sample processing. Cells were permeabilized with 0.5% Triton X-100 in 1 × PBS (pH 7.4), blocked with blocking buffer (4% goat serum in 1 × PBS) and washed with 1 × PBS (pH 7.8). For the click reaction, the cells were incubated with a click reaction mix (200 µM triazole ligand Tris((1-benzyl-1H-1,2,3-triazol-4-yl)methyl)amine (TBTA), 25 µM biotin alkyne tag, 500 µM TCEP and 200 µM $CuSO_4$ in PBS (pH 7.8)) overnight at room temperature, protected from light. The cells were washed with 0.5% Triton X-100 in 1 × PBS (pH 7.4) and PBS and then blocked with blocking buffer. Cells were incubated with a mixture of primary antibodies diluted in blocking buffer (90 min at room temperature). For Lamin-B1 experiment: rabbit-anti-lamin-B1 (1:800; abcam, ref. no.: ab16048), mouse-anti-biotin (1:1,000; Life Technologies, ref. no.: 03–3700), and guinea pig-anti-MAP2 (1:1,000; Synaptic Systems, ref. no.: 188004). For GM130 experiment: mouse-anti-GM130 (1:500; BD Transduction Laboratories, ref. no.: 610823), rabbit-anti-biotin (1:5,000; Cell signaling, ref. no.: 5597) and guinea pig-anti-MAP2 (1:1,000). PLA probes (rabbit PLA-plus and mouse PLA-

minus; Sigma) were applied in 1:10 dilution in blocking buffer supplemented with goat-anti-guinea pig secondary antibody coupled to Alexa Fluor 488 (1:500; Life Technologies) for 1 hr at 37°C in a wet chamber. Cells were then incubated with the ligation reaction mix (Duolink Detection reagents Red, Sigma) for 30 min in a pre-warmed wet chamber at 37°C. Amplification and label probe binding was performed after three washes with wash buffer A with the amplification reaction mixture (Duolink Detection reagents Red, Sigma) in a pre-warmed wet chamber at 37°C for 100 min. After amplification was stopped by washes with wash buffer B (0.2 M Tris, 0.1 M NaCl, pH 7.5), cells were incubated with of 4′,6-diamidino-2-phenylindole (DAPI; 1:1000) for 2 min to stain nuclei. DAPI was removed by washes with wash buffer B and cells were subsequently imaged in wash buffer B.

## Imaging

Cells were selected based on the MAP2, DAPI and PLA signals. Criteria for choosing cells were healthy appearance judged by MAP2 staining (no swelling of dendrites or fragmented pattern) and homogeneous DAPI staining. As the PLA signal was very heterogeneous between different neurons, for each time point the whole dish was inspected and the cells with the overall highest puncta number were selected for each time point. All images for one experiment were acquired within 12 hr after the PLA procedure. Images were acquired with a LSM880 confocal microscope (Zeiss, Oberkochen, Germany) using a 40×/1.4-NA oil objective (Plan Apochromat 40×/1.3 oil DIC UV-IR M27) and a pinhole setting of 29 μm. The lasers were used at 0.8% (488 nm), 0.9% (561 nm) and 2.8% (405 nm) power. Images were acquired in 12-bit mode as z stacks with 2,048 × 2,048–pixel xy resolution through the entire thickness of the cell, pixel dwell times of 0.26 μs and the detector gain in each channel adjusted to cover the full dynamic range but to avoid saturated pixels. Imaging conditions were held constant within an experiment.

## Image data processing and analysis

The number of puncta was assessed using custom built Matlab scripts (*Tuchev, 2017*; code available at https://github.molgen.mpg.de/MPIBR/NeuroBits; copy archived at https://github.com/elifesciences-publications/MPIBR-NeuroBits) and revised manually. Puncta were detected in neuronal somata by finding local intensity maxima in the PLA channel. The mean puncta number per neuron was plotted over time and fitted by a first order exponential function. This function was used to calculate the protein half-life. Representative cell somata were cropped and the puncta channel was thresholded, dilated once and smoothened to improve puncta visualization in figures. The threshold value was kept constant for all images (*Figure 2A*). Puncta numbers for all analyzed neurons are presented in *Figure 2B* and *Figure 2—source data 1*. Ln transformed mean puncta numbers were fit by a linear function and the protein half-life was calculated as described above.

## Acknowledgements

We thank I Bartnik, N Fuerst, A Staab and C Thum for the preparation of primary cell cultures and F. Rupprecht for MS maintenance and assistance with data acquisition. EMS is funded by the Max Planck Society and DFG CRC 1080: Molecular and Cellular Mechanisms of Neural Homeostasis and DFG CRC 902: Molecular Principles of RNA-based Regulation. This project has received funding from the European Research Council (ERC) under the European Union's Horizon 2020 research and innovation programme (grant agreement No 743216).

## Additional information

### Funding

| Funder | Grant reference number | Author |
|---|---|---|
| Horizon 2020 Framework Programme | 743216 | Erin M Schuman |
| Max-Planck-Gesellschaft | Open-access funding | Erin M Schuman |

The funders had no role in study design, data collection and interpretation, or the decision to submit the work for publication.

## Author contributions
Aline R Dörrbaum, Data curation, Formal analysis, Visualization, Writing—review and editing; Lisa Kochen, Formal analysis, Visualization, Writing—review and editing; Julian D Langer, Conceptualization, Data curation, Formal analysis, Supervision, Writing—review and editing; Erin M Schuman, Conceptualization, Supervision, Writing—original draft, Writing—review and editing

## Author ORCIDs
Aline R Dörrbaum 🆔 http://orcid.org/0000-0002-1178-7078
Julian D Langer 🆔 http://orcid.org/0000-0002-5190-577X
Erin M Schuman 🆔 http://orcid.org/0000-0002-7053-1005

## Ethics
Animal experimentation: All experiments were carried out with the approval of the German animal experiment authorities.

## Decision letter and Author response
Decision letter https://doi.org/10.7554/eLife.34202.036
Author response https://doi.org/10.7554/eLife.34202.037

# Additional files

### Supplementary files
• Supplementary file 1. Half-lives, rate constants, standard errors and coefficients of determination for all proteins in all culture types.
DOI: https://doi.org/10.7554/eLife.34202.029

• Supplementary file 2. MS parameters.
DOI: https://doi.org/10.7554/eLife.34202.030

• Supplementary file 3. Sample tables and MaxQuant parameter sets.
DOI: https://doi.org/10.7554/eLife.34202.031

• Transparent reporting form
DOI: https://doi.org/10.7554/eLife.34202.032

### Data availability
All proteomics data associated with this manuscript have been uploaded to the PRIDE online repository (identifier: PXD008596).

The following previously published dataset was used:

| Author(s) | Year | Dataset title | Dataset URL | Database, license, and accessibility information |
| --- | --- | --- | --- | --- |
| Dörrbaum AR, Kochen L, Langer JD, Schuman EM | 2018 | Local and global influences on protein turnover in neurons and glia. | https://www.ebi.ac.uk/pride/archive/projects/PXD008596 | Publicly available at EBI PRIDE (accession no. PXD008596) |

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
