## [Decision Letter]

Thank you for submitting your article "Local and global influences on protein turnover in neurons and glia" for consideration by *eLife*. Your article has been favorably evaluated by a Senior Editor and three reviewers, one of whom is a member of our Board of Reviewing Editors. The following individuals involved in review of your submission have agreed to reveal their identity: Jeffrey N Savas (Reviewer #2); Kelsey Martin (Reviewer #3).

The reviewers have discussed the reviews with one another and the Reviewing Editor has drafted this decision to help you prepare a revised submission.

Summary:

This paper addresses an important unanswered question in studies of neuronal activity-dependent gene expression: what is the turnover rate of neuronal proteins? While many studies have provided insights into stimulus-induced translation, it is difficult to interpret how new translation impacts the neuronal proteome in the absence of information about protein half-life or turnover. While a handful of published studies have provided some insight into protein half-life in neurons, the Schuman lab now directly addresses this question in a systematic manner by quantifying the half-lives of over 5000 proteins in neuronal cultures using a SILAC approach. The strength of the study is from the three independent replicates and the analysis of protein turnover rates.

This provides a resource for future studies, and also reveals the following general principles: 1) proteins in different subcellular compartments have distinct half-lives; 2) proteins within complexes largely have similar half-lives 3) the same proteins have distinct half-lives in glia and neurons; and 4) the presence of glia alters the half-life of proteins in neurons.

However, there are several concerns. Dorrbaum et al. compare combined neuronal/glial cultures with glial-enriched and neuronal-enriched cultures. This is not as straightforward as presented. For example, the addition of Ara-C to the neuronal-enriched cultures is likely to have effects on the neuron in addition to its effects on glia; the number of glia at various time points in the glial-enriched cultures will vary; the mixed and neuron-enriched cultures are from hippocampus while the glial-enriched cultures are from cortex (as shown in xx, different brain regions have different glial populations). To confirm the SILAC results, the authors perform FUNCAT-PLA for 4 candidate proteins. However, they compare SILAC results of mixed cell neuronal/glial cultures with FUNCAT in neurons, even though one of the paper's conclusions is that protein half-life is distinct in different cell types.

Their findings that protein half-life is different in glia and in neurons (Figure 6) raises questions about the analysis of the data from mixed cultures in Figures 1-5. It indicates that it's critical to analyze protein half-life in the most physiologically relevant system. While the authors point out that mixed neuronal cultures "mimic the physiology of cells in-vivo," there are significant differences between neurons in the brain and neurons in culture.

Essential revisions:

To strengthen the manuscript, it is recommended to compare sister neuronal-enriched cultures with and without glial-conditioned medium. This would constitute the same starting material but with distinct extracellular environments. Another way is to examine the proteome – or at least some candidate proteins – with and without activity. This would again allow examination of the same starting material with different "external environments" aka stimuli. All the reviewers agreed these experiments would help to improve the study.

Other comments from the reviewers that should be addressed:

1) Glial cultures are generated from the cortex, but more details are necessary. The relative representation of astrocytes, oligodendrocytes and microglial should be reported. Were oligodendrocytes separated from astrocytes? There is evidence in the literature that glia differ in different brain regions. Shouldn't hippocampal neurons be compared to hippocampal glia?

2) AHA is used to carry out labeling for two hours in a FUNCAT-PLA experiment. Does AHA affect the protein degradation rate, compared to methionine? There is some concern regarding the health of the primary cultures and the incidence of cell death derived from using AHA. An effort should be made to ensure the stability of the AHA-labeled proteins in these experiments.

3) Why did the authors choose to leave 20% of the light medium in the culture when they switched to Heavy? This only complicates the analysis. One alternative could be to use parallel cultures that were heavy labeled and transfer that medium when switching.

4) The disconnect between the FUNCAT-PLA data and MS results is surprising. The authors are world leaders on this and clearly acknowledge and comment on the discrepancy but the reviewer is wondering which measure is more accurate? Its good the trend were the same.

5) Overall, this manuscript is a very nice analysis, the figures are well made, and the text is generally very well written. The biggest problem for this reviewer was that there were no major conclusions reached. The analysis is systematic but a punch line is absent and the current version seems to reach the conclusion that overall – "protein degradation is complicated in cultured neurons." The novelty is not particularly high and the most interesting findings were in attempted to be addressed in Figure 6 however they were a bit too weak. Overall, this reviewer found this manuscript to be well constructed but left me a bit unsatisfied and wanting more.

6) Figure 4 is not particularly informative or easy to interpret and conclusions regarding the glutamate receptors while apparently significant do not come across as compelling. Can this data be presented in a different way?

7) Figure 5 appears to be a rather arbitrary collection of protein complexes and again the conclusions are not particularly firm or clear.

8) The legend to Figure 6 is confusing and does not match the text. Figure 6A is surprising, since the mixed and neuron enriched traces are similar. PCA in Figure 6—figure supplement 1H is not convincing and should be improved. The title does not accurately describe the manuscript and only deals with the last figure. Overall the analysis in Figure 6 is preliminary and the effect of cell division needs more attention.

---

## [Author Response]

Essential revisions:To strengthen the manuscript, it is recommended to compare sister neuronal-enriched cultures with and without glial-conditioned medium. This would constitute the same starting material but with distinct extracellular environments. Another way is to examine the proteome – or at least some candidate proteins – with and without activity. This would again allow examination of the same starting material with different "external environments" aka stimuli. All the reviewers agreed these experiments would help to improve the study.

We have conducted the requested experiment: we compared protein turnover rates of sister neuronal-enriched cultures with and without glial-conditioned medium (see Materials and methods for details). As shown in Figure 6I of the revised manuscript, there is a very high correlation between protein turnover in neuron-enriched cultures with and without conditioned medium. Only 29 proteins showed significantly different turnover rates in the different conditions. Of those, 13 proteins show a faster turnover in the presence of conditioned medium and 16 show a faster turnover without conditioned medium. Though significant, the observed differences in protein turnover are small with an average difference in relative protein decay of 5% after 7 days of SILAC treatment (Figure 6—source data 4). Also those proteins that displayed a different half-live in mixed versus neuron-enriched cultures do not show different half-lives in the comparison of neuron-enriched cultures with and without conditioned medium. We therefore conclude that the observed differences in protein turnover rates in the mixed versus neuron-enriched cultures are a result of cellular interactions, while the composition of the medium does not exert a major effect on protein turnover rates.

We also agree with the reviewers that it is very interesting to examine protein regulation in neurons with activity. Indeed we conducted (and recently published Schanzenbächer et al., 2016 and Schanzenbächer et al., 2018) experiments examining newly synthesized proteins following either increases or decreases in activity. In these published studies, we found that specific sets of proteins show regulated synthesis rates during the stimulus compared to untreated control samples. It will also be interesting to examine the regulation of protein turnover with activity manipulations, but this will require at least a year of additional work.

Other comments from the reviewers that should be addressed:1) Glial cultures are generated from the cortex, but more details are necessary. The relative representation of astrocytes, oligodendrocytes and microglial should be reported. Were oligodendrocytes separated from astrocytes? There is evidence in the literature that glia differ in different brain regions. Shouldn't hippocampal neurons be compared to hippocampal glia?

To assess the cellular composition of the glia-enriched cultures, we compared relative protein intensities of cell type specific marker proteins in glia-enriched and mixed cultures using our Mass Spec data. As described in the manuscript, neuronal marker proteins were significantly de-enriched in glia-enriched cultures compared to mixed cultures (Figure 6— – figure supplement 1C). To determine the relative representation of different glia cell types, we extracted the top 20 most enriched proteins in cultured astrocytes, oligodendrocytes and microglia reported in the proteomic studies of Sharma et al. (2015), which were also present in our data set, and compared their relative intensities between our mixed cultures and glia-enriched cultures. A higher number of astrocyte and microglia marker proteins were detected in the glia-enriched cultures compared to mixed cultures (20 vs. 12 astrocyte marker in glia-enriched and mixed cultures, respectively, and 19 vs. 10 microglia marker in glia-enriched and mixed cultures, respectively; see Figure 6—figure supplement 1D). In addition, most of the astrocyte and microglia markers that were detected in both culture types showed higher relative intensities in glia-enriched cultures compared to mixed cultures indicating that astrocytes and microglia are highly enriched in glia-enriched cultures compared to mixed cultures. For oligodendrocytes, a similar number of marker proteins were detected in glia-enriched (17) and mixed cultures (18). Of those oligodendrocyte markers that were detected in both culture types, 8 showed higher relative intensities in glia-enriched cultures, while 4 showed higher relative intensities in mixed cultures. These findings indicate that our glia-enriched cultures are mainly enriched in astrocytes and microglia, while there is a similar fraction of oligodendrocytes in glia-enriched and mixed cultures. This new analysis is now also included in the revised manuscript (Figure 6—figure supplement 1D-E).

To investigate cellular differences of cultured glia cells obtained from cortex and hippocampus, we prepared glia-enriched cultures from cortex and hippocampus and compared their proteomes. 97% of all identified proteins were detected in both glia cultures and their relative expression levels were highly correlated (Pearson correlation ≥ 0.95; see Figure 6—figure supplement 4). These results indicate that primary glia cultures prepared from hippocampus and cortex are very similar. We include this new experiment in our revised manuscript (Figure 6—figure supplement 4).

2) AHA is used to carry out labeling for two hours in a FUNCAT-PLA experiment. Does AHA affect the protein degradation rate, compared to methionine? There is some concern regarding the health of the primary cultures and the incidence of cell death derived from using AHA. An effort should be made to ensure the stability of the AHA-labeled proteins in these experiments.

Previous studies from our lab demonstrated that AHA incorporation for 2 hours is not toxic for cells and does not affect global protein synthesis and degradation rates (Dieterich et al., 2006). We thank the reviewers for mentioning this point and now reference these findings in the revised manuscript. In the present study, the neurons were visually inspected under the microscope and the majority of neurons showed a healthy appearance; importantly there was no difference in viability or morphology between AHA treated cells and cells in methionine control samples. For the FUNCAT-PLA experiment, only neurons with robust, elaborate dendritic trees characteristic of 18-26 DIV in our lab were used.

3) Why did the authors choose to leave 20% of the light medium in the culture when they switched to Heavy? This only complicates the analysis. One alternative could be to use parallel cultures that were heavy labeled and transfer that medium when switching.

During the medium change from light to heavy SILAC medium, a thin layer of light medium (~1 mL) remained on the cells and an excess of “heavy” medium was added resulting in a ratio of heavy-to-light medium of 4:1. In our pilot studies, we found that this partial medium change is a robust procedure that resulted in better viability of the cells. This is especially true for neuron-enriched cultures, which are particularly sensitive to stress. We note that other dynamic SILAC studies in primary neuronal cultures also avoided complete medium changes (e.g. Cohen et al., 2013 and Mathieson et al., 2018). We also tested complete medium changes using “light” and “heavy” labelled sister cultures, but in our hands, this procedure was less robust and less reproducible compared to the 80/20 medium change. In order to determine and to correct for the rate of incorporation of “light” arginine and lysine into newly synthesized proteins after the medium change, we used peptides with one missed tryptic cleavage site that contain two arginine and/or lysine residues. We discuss this in our Materials and methods section. We agree that this procedure may complicate the data analysis, but we are convinced that this is justifiable with regard to the improved cell viability and experimental robustness of the data.

4) The disconnect between the FUNCAT-PLA data and MS results is surprising. The authors are world leaders on this and clearly acknowledge and comment on the discrepancy but the reviewer is wondering which measure is more accurate? Its good the trend were the same.

As described in the manuscript, the discrepancy might be explained by an increased sensitivity of the pulse-chase experiment to detect the rapid degradation of misfolded proteins, which might be missed by the dynamic SILAC approach. For each protein there might be a stable/functional sub-population and an unstable sub-population that is degraded shortly after synthesis. Due to their fast degradation, these unstable proteins are of low abundance in the cell lysate and thus have little contribution to the measured MS signal. The turnover rates obtained by the dynamic SILAC approach hence mainly describe the turnover of the stable protein sub-population and neglect the unstable sub-population. In contrast, all newly synthesized proteins (stable and unstable) are visualized in the FUNCAT-PLA experiment and the stable and unstable sub-populations contribute equally to the results.

5) Overall, this manuscript is a very nice analysis, the figures are well made, and the text is generally very well written. The biggest problem for this reviewer was that there were no major conclusions reached. The analysis is systematic but a punch line is absent and the current version seems to reach the conclusion that overall – "protein degradation is complicated in cultured neurons." The novelty is not particularly high and the most interesting findings were in attempted to be addressed in Figure 6 however they were a bit too weak. Overall, this reviewer found this manuscript to be well constructed but left me a bit unsatisfied and wanting more.

We appreciate both the compliments and the critical comments of the reviewer. Regarding the criticism, we take responsibility for not convincingly demonstrating the major conclusions reached. We summarize here the key points:

- There is a wide distribution of protein half-lives in primary neuronal cultures with half-lives ranging from < 1day to > 20 days.

- Proteins at distinct cellular localizations have different half-lives compared to the entire population (e.g. mitochondrial proteins show longer half-lives whereas membrane proteins show shorter half-lives). Despite their remote location, synaptic proteins as a group do not show significantly different half-lives compared to the entire population.

- In primary hippocampal cultures, protein half-lives do also correlate with protein functions. Distinct functional groups, e.g. receptors and signaling molecules are enriched in the fraction of short-lived proteins, while others, such as histones and proteins involved in the energy metabolism are enriched in the fraction of long-lived proteins.

- Multi-protein complex members display more similar half-lives compared to randomly selected proteins. Still, for several complexes, half-lives of the constituents show considerable variability. Our data suggests that the half-life distribution within complexes correlates with the assembly and disassembly dynamics of the complexes.

- Different cell types of the brain have different protein turnover rates.

- Protein half-lives are not strictly cell-type specific, but also depend on cellular interactions.

6) Figure 4 is not particularly informative or easy to interpret and conclusions regarding the glutamate receptors while apparently significant do not come across as compelling. Can this data be presented in a different way?

Interestingly, we found that, despite their remote location, synaptic proteins do not have significantly different half-lives compared to non-synaptic proteins. Furthermore, as synaptic proteins show heterogeneous protein half-lives as a group we aimed to depict the distribution and heterogeneity of these protein half-lives in this figure. Figure 4 displays all synaptic proteins found in our dataset and we organized the proteins according to their localization in the pre- or post-synapse, respectively, although they might not be exclusively found at that location. We now further improved the figure to better indicate interacting proteins (e.g. CamK2), proteins that have similar functions (e.g. scaffold proteins of the post-synaptic density) and protein involved in different steps of the synaptic vesicle cycle. We chose this representation as it enables the reader to easily spot the dynamics of interacting and functionally related proteins within the synapse. For instance, Figure 4 illustrates that glutamate receptors as a group have clearly shorter half-lives than the average of the population and proteins associated with synaptic vesicles have clearly longer half-lives.

7) Figure 5 appears to be a rather arbitrary collection of protein complexes and again the conclusions are not particularly firm or clear.

We apologize that the selection of the complexes was not described in more detail in the manuscript. There is emerging interest in both cell biology and neurobiology on the behavior of “macromolecular complexes” or proteins that carry out their cellular function as part of multi-subunit machines. Our analysis of 314 protein complexes (from CORUM database) revealed that proteins that are members of stable complexes have more similar half-lives compared to randomly selected proteins (Figure 5A). However, for some complexes, the half-lives of the constituents showed substantial deviation – suggesting that these complexes might be subject to dynamic exchange of new and old protein subunits. This variability of protein half-lives within complexes does not correlate with the number of constituents in a complex (Figure 5—figure supplement 1). We next investigated if a protein’s half-life correlates with its position within the complex and if the half-life variation within a complex correlates with its assembly and disassembly dynamics. For this means, we selected multi-protein complexes with different structural properties and different assembly and disassembly dynamics. The spliceosome is a very dynamic complex that assembles and disassembles for each splicing event. In contrast, the ribosome is thought to be a rather static complex. After their assembly in the nucleus and nucleolus, the large and small ribosomal subunits are released into the cytoplasm where they act as protein synthesis “machines”. Though the small and large subunit can attach and detach, the subunits themselves are considered to be stable throughout their “life time”. The heterogeneity of half-lives observed for the ribosomal proteins suggest that this stability for life might be reconsidered. The ATP synthase is thought to assemble in a step-wise manner. The membrane-embedded rotor as well as the soluble stalk and head domains are produced and assembled independently. Full ATP synthase assembly is a finely-tuned process dependent on the energy demand of the cell. We observed similar half-lives of the ATP synthase subunits. The nuclear pore complex is a very large membrane-spanning protein complex that contains static as well as dynamic regions. Our data suggests that there is a correlation between the dynamic behavior of a complex and the half-lives of its constituents. Based on this observation, we propose a concept that half-lives of protein complex constituents can be used to better understand the dynamics of protein complexes that are not yet well studied. We worked on this concept in the revised manuscript to better explain the selection of the protein complexes and the aim of the analysis.

8) The legend to Figure 6 is confusing and does not match the text.

We apologize if Figure 6 legend was confusing. We improved Figure 6 and the corresponding figure legend in the revised manuscript.

Figure 6A is surprising, since the mixed and neuron enriched traces are similar.

We modified Figure 6 and now present the protein half-life distributions of the different culture types in histograms (Figure 6B in the revised manuscript). Please note that this panel does not show a protein-wise comparison. Though, there is an overall similar distribution of protein half-lives in the mixed and neuron-enriched cultures, importantly, individual proteins can still exhibit different turnover rates in the different culture types, as shown in the protein-wise comparison in Figure 6D.

PCA in Figure 6—figure supplement 1H is not convincing and should be improved.

To better point out differences in the relative protein expression levels and to improve the PCA analysis, we log_2_ transformed normalized protein intensities and used these values for the PCA analysis. Note that only proteins that were quantified in all culture types could be used for the PCA analysis.

The title does not accurately describe the manuscript and only deals with the last figure.

We have clarified the wording of our title in the Discussion. We believe it is appropriate.

Overall the analysis in Figure 6 is preliminary and the effect of cell division needs more attention.

The aim of our study is to monitor protein turnover under steady state conditions, while protein synthesis and degradation are in a balance. We agree with the reviewers that it is crucial to consider cell division. During cell proliferation more proteins are produced and hence there is a misbalance between protein synthesis and degradation resulting in underestimation of protein half-lives. Neurons are post-mitotic cells that do not undergo cell division, whereas glia cells are mitotic cells that are able to divide. At the time of our SILAC experiments (starting at DIV 18-19), the cells (neurons and glia) formed confluent monolayers that lacked overt indicators of growth. In response to the reviewer’s concern, we estimated glial cell division over the time course of the experiment; to do this, we monitored the temporal expression levels of 43 glia markers in mixed cultures (Figure 6—figure supplement 5A). Increasing expression levels of glia markers would indicate cell division. Though different markers show different expression profiles over time, there is no systematic increase of glia marker proteins over the time course of the experiment, suggesting that there is minimum cell division in our mixed cultures. As an additional measure of cell division, we investigated the half-life of the cell division-dependent protein Histone H3.1. As described in the manuscript, the heavy isotopically labelled form of Histone H3.1 is only produced by cell division and not by protein turnover in post-mitotic cells. During the life-time of post-mitotic cells, Histone H3.1 can be degraded and replaced by different histone variants. In mixed cultures, Histone H3.1 shows a comparably long half-life of 11.6 days (Figure 6—figure supplement 5B) indicating that there is a low rate of ongoing glia cell division in mixed cultures. As mentioned above, cell division can lead to an underestimation of protein half-lives (calculated protein half-life is shorter than real protein half-life) for proteins that are produced in the dividing cell type. This underestimation will be more pronounced for extremely long-lived proteins compared to short-lived proteins. We next used the half-life of Histone H3.1 to compare the rate of cell division in the different culture types. Similar half-lives for Histone H3.1 were obtained in mixed cultures (11.6 days) and glia-enriched cultures (11.2 days) demonstrating that a similar low rate of cell division takes place in both culture types. Consequently, cell division cannot account for the systematically shorter protein half-lives in glia-enriched cultures compared to mixed cultures. The longest half-life for Histone H3.1 was observed in neuron-enriched cultures (> 30 days) consistent with the observation that mitotic cells are strongly de-enriched in this culture type. We rephrased and added to the revised manuscript a discussion of the observed rate of cell division and its effects on half-life determination. We further added the analysis of the temporal expression levels of glia markers in mixed cultures, as well as the analysis of Histone H3.1 turnover as a new supplementary figure (Figure 6—figure supplement 5).